# Genome-wide association study of cerebellar volume provides insights into heritable mechanisms underlying brain development and mental health

Elleke Tissink [1], Siemon C. de Lange [1,2], Jeanne E. Savage [1], Douglas P. Wightman [1], Christiaan A. de Leeuw [1], Kristen M. Kelly[3], Mats Nagel [1], Martijn P. van den Heuvel [1,4] & Danielle Posthuma [1,4✉]

Cerebellar volume is highly heritable and associated with neurodevelopmental and neuro-degenerative disorders. Understanding the genetic architecture of cerebellar volume may improve our insight into these disorders. This study aims to investigate the convergence of cerebellar volume genetic associations in close detail. A genome-wide associations study for cerebellar volume was performed in a discovery sample of 27,486 individuals from UK Biobank, resulting in 30 genome-wide significant loci and a SNP heritability of 39.82%. We pinpoint the likely causal variants and those that have effects on amino acid sequence or cerebellar gene-expression. Additionally, 85 genome-wide significant genes were detected and tested for convergence onto biological pathways, cerebellar cell types, human evolutionary genes or developmental stages. Local genetic correlations between cerebellar volume and neurodevelopmental and neurodegenerative disorders reveal shared loci with Parkinson's disease, Alzheimer's disease and schizophrenia. These results provide insights into the heritable mechanisms that contribute to developing a brain structure important for cognitive functioning and mental health.

[1] Department of Complex Trait Genetics, Center for Neurogenomics and Cognitive Research, Vrije Universiteit Amsterdam, Amsterdam Neuroscience, 1081 HV Amsterdam, The Netherlands. [2] Department of Sleep and Cognition, Netherlands Institute for Neuroscience, an institute of the Royal Netherlands Academy of Arts and Sciences, Amsterdam, The Netherlands. [3] Institute for Behavioral Genetics, University of Colorado Boulder, Boulder, CO, USA. [4] Department of Child and Adolescent Psychiatry, Section Complex Trait Genetics, Amsterdam Neuroscience, Vrije Universiteit Medical Center, Amsterdam UMC, Amsterdam, The Netherlands. ✉email: danielle.posthuma@vu.nl

nsights from the neuroscientific field have changed the perception that the cerebellum is predominantly involved in somatic motor control and the triad of clinical ataxias[1]. The cerebellum is now widely considered as a major centre for multiple cognitive functions such as attention[2] and verbal working memory[3]. The association between structural abnormalities and neurodevelopmental and neurodegenerative disorders[4] makes investigating the biology of the cerebellum an essential area of neuroscience research. Twin studies have estimated the heritability of cerebellar volume at 88%[5]. However, it remains unclear which and specifically how genetic factors are responsible for interindividual variability in cerebellar volume and relate to genetic factors involved in neurodevelopmental and neurodegenerative disorders.

One way to examine the genetic architecture of neuroimaging-derived phenotypes is to evaluate common genetic variants (single-nucleotide polymorphisms or SNPs) for association in a genome-wide association study (GWAS)[6,7]. Based on these SNP effects, cerebellar volume heritability ($h^2_{SNP}$) has been estimated at 45.3–46.8% and 21 genes have been identified for potential follow-up[8]. However, a large body of research has shown that methods that exploit the polygenic signal of traits to look for convergence beyond genes onto biological pathways, cell types or developmental time windows have the potential to provide more meaningful starting points for follow-up experiments[9]. In parallel, methods that zoom in on a locus level to examine the local genetic overlap between traits or pinpoint the likely causal variant(s) facilitate the prioritisation of SNPs or genes for future studies[9]. Therefore, a thorough interrogation of cerebellar volume at the level of genetic variants with detailed follow-up focused on translating genetic loci into mechanistic hypotheses could elucidate why cerebellar volume varies between individuals and contributes to neurodevelopmental and neurodegenerative disease.

We perform a GWAS on UK Biobank data of total cerebellar grey and white matter volume (discovery $N = 27,486$, replication $N = 3906$) to assess which common genetic variants and genes contribute specifically to cerebellar volume. We annotate and finemap discovered loci to pinpoint the most impactful and likely causal variants and describe the genetic architecture of cerebellar volume by its polygenicity and discoverability compared to cerebral and subcortical volume. It is examined to what extent cerebellar volume-specific genes converge onto biological pathways or specific cell types located in the cerebellum. We further investigate whether these genes display a specific temporal expression pattern through development or associate with genes important for human evolution. Supplementary Notes 1 and 2 describe how this approach differs from previous endeavours. The overlap between the genetic profile of cerebellar volume and neurodevelopmental and neurodegenerative disorders on a global and regional level highlights the impact of cerebellar volume genetic variation on ADHD, schizophrenia, Alzheimer's and Parkinson's disease. Our results provide insights into the heritable mechanisms that contribute to developing a key brain structure for cognitive functioning and mental health.

## Results

### The SNP-based heritability of cerebellar volume is 39%: 30 genomic loci identified.
GWAS of cerebellar volume in 27,486 individuals assessing 9,380,224 SNPs identified 1,789 genome-wide significant (GWS) SNPs ($P < 5 \times 10^{-8}$) located in 30 distinct genomic loci (Fig. 1a; loci comparison with previous GWAS[8] in Supplementary Data 19). SNP-based heritability ($h^2_{SNP}$) equalled 39.8 % (SE = 3.14%). The inflation of the GWAS-derived test statistics ($\lambda_{GC} = 1.18$, mean $\chi^2 = 1.24$; Fig. 1b) could be accounted for by polygenicity since the LDSC intercept of 1.03 (SE = 0.008) indicated

negligible confounding bias. We continued by partitioning the $h^2_{SNP}$ into functional genomic categories to test for enrichment of heritability in these categories (Fig. 1d). With partitioned LD Score regression (LDSC) we used the associations of all SNPs, because much of the $h^2_{SNP}$ of polygenic traits lies in SNPs that do not reach genome-wide significance[10]. An enrichment of $h^2_{SNP}$ after Bonferroni correction was observed in four functional genomic categories, specifically within regions with three distinct types of chromatin modifications (Supplementary Data 1). The first category, genomic regions with H3K4me1 peaks (enrichment = 4.45, SE = 0.93, $P = 1.45 \times 10^{-4}$), is often used to distinguish enhancers from promotors[11]. In particular active enhancers are associated with additional H3K27ac[12]. Genomic regions with this H3K27ac modification (Hnisz[13]; enrichment = 1.98, SE = 0.18, $P = 1.32 \times 10^{-7}$) and super-enhancers that were tagged using H3K27ac in ref. [13] (enrichment = 2.20, SE = 0.22, $P = 4.96 \times 10^{-7}$) showed significant enrichment of cerebellar volume $h^2_{SNP}$ as well. Super-enhancers are large regions of regulatory elements that can affect cell type-specific and tissue-specific transcription[14]. Genomic regions with H3K9ac peaks, often active gene promoters[15], were enriched of cerebellar volume $h^2_{SNP}$ (enrichment = 10.91, SE = 2.76, $P = 2.60 \times 10^{-4}$). Regions with these chromatin modifications show highly dynamic chromatin accessibility during both early and postmitotic stages of cerebellar granule neuronal maturation[16]. This form of chromatin plasticity is suggested to contribute to establishing synapses and, in adults, even to transcription-dependent forms of learning and memory[16].

We estimated three metrics to describe the genetic architecture of total cerebellar volume using univariate MiXeR[17] (see Methods) and compared these to two other major volumes in the brain, namely total cerebral and total subcortical volume (Supplementary Data 2). Cerebellar ($M = 4.36 \times 10^{-4}$, SD = $4.12 \times 10^{-5}$) and cerebral volume ($M = 3.96 \times 10^{-4}$, SD = $5.99 \times 10^{-5}$) appeared to show lower polygenicity ($\pi$; proportion of independent causal SNPs) then subcortical volume ($M = 7.22 \times 10^{-4}$, SD = $5.24 \times 10^{-5}$). As less polygenic traits tend to have more causal SNPs with larger effect sizes, cerebellar volume showed the highest discoverability (variance of effect sizes of causal SNPs) among the three major brain volumes ($\sigma^2_\beta$ $M = 3.60 \times 10^{-4}$, SD = $2.52 \times 10^{-5}$), meaning that it has on average stronger effects. This is also illustrated by Supplementary Fig. 1, which shows that cerebellar volume shows the highest squared standardised effect sizes for independent significant SNPs with low minor allele frequencies (MAF). All metrics for cerebellar, cerebral and subcortical volume can be found in Supplementary Data 2. Genetic correlations between these three major brain volumes are described in Supplementary Note 3 and Supplementary Data 20 and 21.

### Genomic location and functions of candidate cerebellar volume SNPs.
We used FUMA v1.3.6a[18] to annotate variants in associated loci based on available information about regional LD patterns and functional consequences of variants. FUMA categorised 3,611 SNPs with (1) $r^2 \geq 0.6$ with one of the GWS SNPs, (2) a suggestive $P$ value ($<1 \times 10^{-5}$) and (3) a minor allele frequency (MAF) >0.005, as candidate SNPs. In concordance with our partitioned $h^2_{SNP}$ results, we observed a Bonferroni-significant enrichment of candidate SNPs in brain-specific chromatin states six (genic enhancers; OR = 2.66, $P = 1.59 \times 10^{-4}$) and seven (enhancers; OR = 1.51, $P = 9.14 \times 10^{-6}$) that are both tagged by H3K4me1 modifications[19] (for all enrichment results, see Supplementary Data 3 and Fig. 1c).

As is observed in other complex traits[20], the majority of candidate SNPs were located in intronic (57.84%, OR = 1.59, $P = 2.20 \times 10^{-149}$) and intergenic regions (29.89%, OR = 0.64,

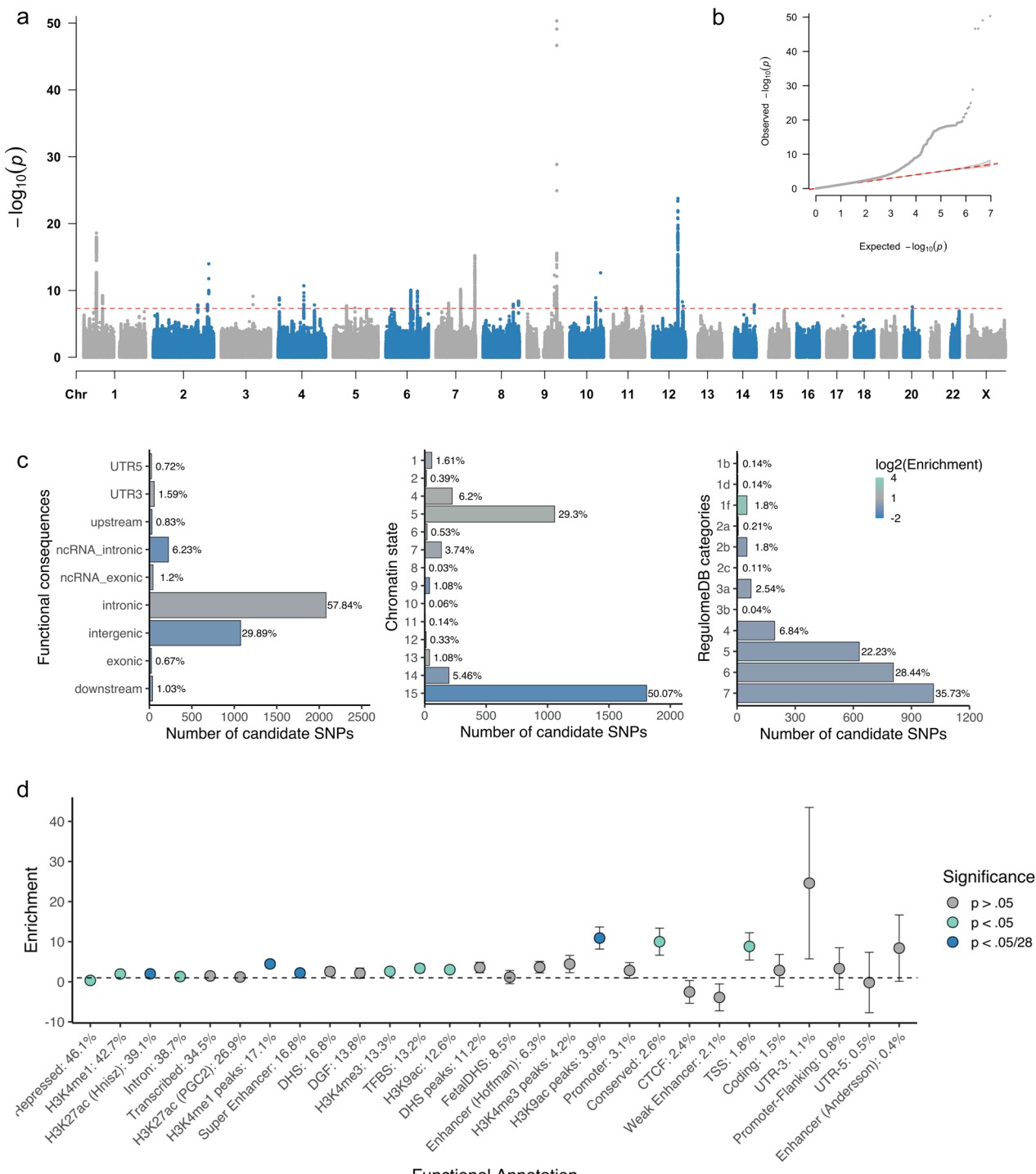

**Fig. 1 SNP-based GWAS of cerebellar volume identifies 30 genome-wide significant loci. a** Manhattan plot of −log10(P) values from GWAS for cerebellar volume. **b** Quantile–quantile (QQ) plot of −log10(P) values from GWAS of cerebellar volume; **c** Proportion of candidate SNPs with corresponding genomic location, RegulomeDB score and brain-specific common chromatin state as assigned by FUMA. Log2(Enrichment) values are colour-coded (depletion coloured in blue), and corresponding P values are available in Supplementary Data 3. **d** Enrichment (and standard errors) of cerebellar volume heritability in functional genomic categories, also available in Supplementary Data 1.

$P = 1.01 \times 10^{-92}$), which complicates functional interpretation (Fig. 1c). Yet four candidate SNPs were nonsynonymous exonic SNPs (ExNS; rs1060105, rs2234675, rs13107325, rs6962772) in the *SBNO1*, *PAX3*, *SLC39A8* and *ZNF789* genes respectively, of which the latter three ExNS reached genome-wide significance (Supplementary Data 4). *PAX3* encodes for a transcription factor that is involved in the neural tube and neural crest development[21]

and is known to induce axonal growth and neosynaptogenesis in the mammalian olivocerebellar tract[22]. The missense variant in exon 6 leads to a threonine to lysine change and each effect allele accounts for a decrease in cerebellar volume of 0.17 SD ($\beta = -1688.74$, SE = 218,27). The exonic variant in the *SLC39A8* gene causes an alanine to threonine change in the zinc transporter ZIP8 the gene encodes for. Each effect allele relates to a 0.11 SD

increase in cerebellar volume ($\beta = 1157.91$, SE = 172.63). The effect alleles of rs6962772 (threonine >alanine) and rs1060105 (salcatonin > asparagine) correspond to a 0.07 SD ($\beta = 750.43$, SE = 122.01) and 0.05 SD ($\beta = 548.25$, SE = 109.13) increase, respectively. The corresponding CADD scores of these four ExNS SNPs (22.9, 25.2, 23.1, 12.6, respectively) indicated high deleteriousness: a property representing reduced organismal fitness, strongly correlating with molecular functionality and pathogenicity[23].

All cerebellar volume candidate SNPs were also significantly enriched for low RegulomeDB scores (Fig. 1b) (1b, OR = 15.74, $P = 1.43 \times 10^{-4}$; 1d, OR = 17.34, $P = 9.87 \times 10^{-5}$; 1f, OR = 18.86, $P = 7.76 \times 10^{-46}$). Low scores represent high confidence that these SNPs have functional consequences, based on known associations with gene expression levels and the likelihood to affect transcription factor binding[24]. Supplementary Data 5 provides a detailed overview of the functional impact of all variants in the identified genomic loci.

FUMA detected 37 lead SNPs (Supplementary Data 6), representing independent variants most strongly associated with cerebellar volume. However, lead SNPs are not necessarily causal[25] and could simply be correlated with the true causal SNP that was not measured on the microarray. Statistical fine-mapping can help to identify the underlying causal SNP(s), hence we applied the Bayesian fine-mapping tool FINEMAP[26] to the 29 loci with a median number of variants of 388 per locus. The median number of variants in the 95% credible sets was 233 per locus. We used a stringent posterior inclusion probability (PIP) threshold of >95% resulting in the selection of four credible causal SNPS (rs111891989, rs118017926, rs72754248, rs2234675) in four distinct loci. These SNPs had the greatest possibility to explain the association with cerebellar volume and overlapped with four lead SNPs identified in FUMA. The intronic variant rs72754248 in the *PAPPA* gene was the variant with the strongest association in the cerebellar volume GWAS ($P = 4.73 \times 10^{-51}$). *PAPPA* encodes for the pregnancy-associated plasma protein A, which can cleave insulin-like growth factor binding proteins (IGFBP) to regulate IGF1 availability. rs111891989 is located in an intron of *MSI1*, that encodes for an RNA-binding protein which binds to ROBO3 to regulate the midline crossing of pre-cerebellar neurons[27]. rs118017926 is an intronic variant, located in the *ZNF462* gene that encodes for zinc-finger transcription factor important for embryonic neurodevelopment[28]. The fourth credible rs2234675 corresponds to the ExNS SNP in *PAX3* that was discussed above. Supplementary Data 7 includes the characteristics of all SNPs from the 95% credible sets. Supplementary Fig. 2 includes LocusZoom plots for the top three significant loci, indicating FINEMAP credible SNPs, lead SNPs and eQTLs.

**Top genes implicated in cerebellar volume suggest role for IGF1 regulation.** While associated variants were mapped to 189 genes based on genomic position (located <10 kb from or within a gene (Supplementary Data 8), we also used the full GWAS results to conduct a gene-based analysis using MAGMA (Fig. 2). This resulted in 85 significant genes ($P < 0.05/18,852$ genes tested) associated with cerebellar volume (Supplementary Data 9). In total, 61 of these were also part of the 189 that were implicated because of the presence of a GWS SNP. The five most strongly associated genes with cerebellar volume included *PAPPA* ($P = 3.37 \times 10^{-44}$, in line with our SNP results discussed above), *WASHC3* ($P = 2.50 \times 10^{-27}$), *PARPBP* ($P = 2.5151e\text{-}25 \times 10^{-25}$), *IGF1* ($P = 6.85 \times 10^{-23}$) and *C1orf185* ($P = 3.51 \times 10^{-22}$). We also linked GWS SNPs to genes based on whether the SNP was known to be associated with the cerebellum-specific expression of that gene (expression quantitative trait loci; eQTL). This resulted

in the mapping of 1197 SNPs to 32 genes for which eQTL associations in cerebellar tissue were available (Supplementary Data 8). Nineteen of these eQTL cerebellum genes overlapped with the positional strategies mentioned above, 13 were additionally discovered.

**Looking for convergence of cerebellar volume genes on a wide range of gene sets.** Next, we tested whether cerebellar volume-gene associations showed a relationship with age-specific gene expression. For this purpose, fetal, infant, adolescent and adult RNA sequencing data from cerebellar cortex tissue samples of 25 different ages from the Brainspan database[29,30] were examined. MAGMA gene property analysis showed that the stronger genes were expressed in postconceptional week 17 ($\beta = 0.018$, SE = 0.009, $P = 0.027$) and 35 ($\beta = 0.029$, SE = 0.013, $P = 0.015$), the stronger their genetic association with cerebellar volume. These associations did not survive Bonferroni correction (Supplementary Data 10), but a general difference between the effect direction of pre- and post-natal gene expression timepoints was apparent from Supplementary Fig. 3b. We used a fixed-effects model to validate this observation ("Methods") and detected a highly significant difference ($P = 9.43 \times 10^{-5}$) in mean effect between prenatal ($\mu_{pre} = 0.015$) and post-natal ($\mu_{post} = -0.018$) gene expression timepoints. This indicates that, although there is no Bonferroni-significant involvement found for the gene expression at any one-time point, on average the effect of pre-natal gene expression is stronger than that of post-natal gene expression.

We then investigated whether associated genes converged on biological pathways, an evolutionary signature, or cerebellum-specific cell types. To this end, we conducted gene-set analyses in MAGMA on the gene-based summary statistics using 7246 MSigDB[31] gene sets, one gene set including genes in human accelerated regions (HARs) that are highly different between humans and other species[32], and a cell type-specificity analysis in FUMA using cerebellar cell types from the DropViz Level 1 database[33,34]. No evidence was found for enrichment in any of the tested MSigDB gene sets (Supplementary Data 11) or the HAR gene set ($\beta = 9.88 \times 10^{-3}$, SE = $2.75 \times 10^{-2}$, $P = 0.36$). Cell type-specificity analysis for nine murine cell types in cerebellar tissue resulted in a nominal significant association for the enrichment of associated genes in astrocytes ($\beta = 0.01$, SE = 0.006, $P = 0.029$), yet this association did not survive the Bonferroni correction of $\alpha = (0.05/9 = )\ 0.056$ (see Supplementary Fig. 3a and Supplementary Data 12).

**Global and local genetic overlap between cerebellar volume and disease.** Neuroimaging studies suggest substantial evidence for phenotypic correlations between cerebellar volume and neurodevelopmental disorders[35], such as attention deficit hyperactivity disorder (ADHD), autism spectrum disorder (ASD) and schizophrenia (SCZ), as well as neurodegenerative disorders, such as Parkinson's disease (PD)[36] and Alzheimer's disease (AD)[37]. Here, we tested whether global genetic correlations ($r_g$) between cerebellar volume on the one hand and SCZ, ASD, ADHD, PD, and AD, on the other hand, were significantly different from zero using LD Score Regression (Supplementary Data 13). We did not find any statistically significant global $r_g$ between cerebellar volume and individual disorders after correction for multiple testing (Fig. 3).

Since global $r_g$'s are an average of local correlations across the genome, there is the possibility that substantial, but contrasting local $r_g$'s are masked. To test whether this was the case, we computed local $r_g$ in SUPERGNOVA (Fig. 4 and Supplementary Data 14). Although the global $r_g$ between cerebellar volume and PD was close to zero ($r_g = 0.08$, SE = 0.05, $P = 0.089$), we identified a locus on chromosome 14 that did show significant local $r_g$ with cerebellar volume (Table 1) but did not reach GWS

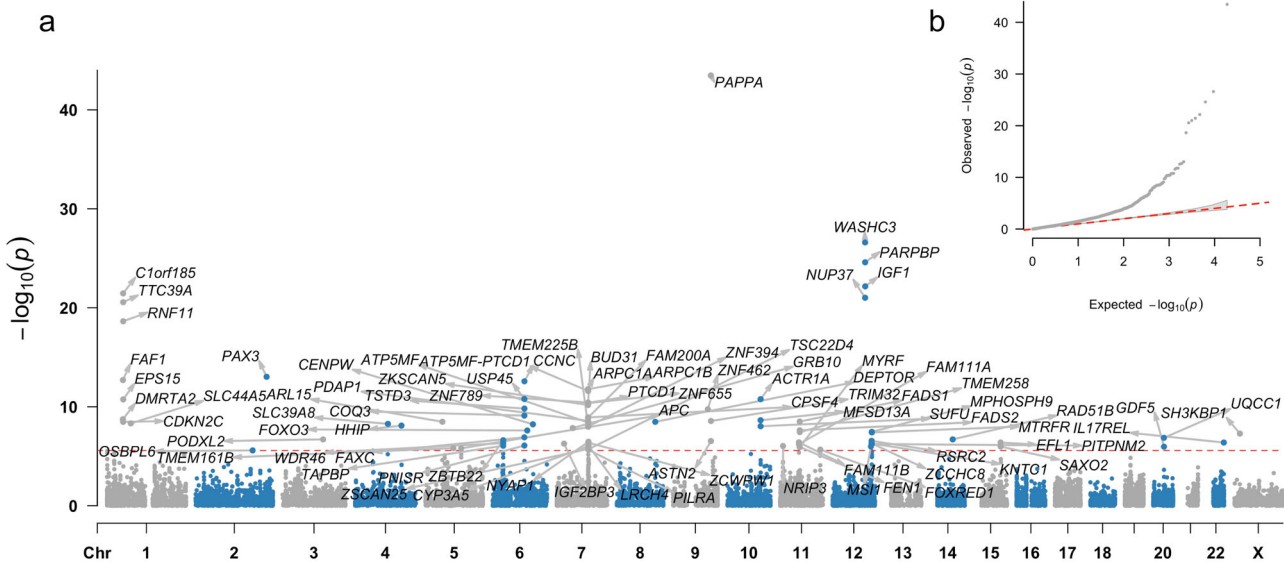

**Fig. 2 Gene-based GWAS of cerebellar volume identifies 85 genome-wide significant genes. a** Manhattan plot of −log10(P) values from the gene-based GWAS for cerebellar volume in MAGMA. **b** Quantile–quantile (QQ) plot of −log10(P) values from gene-based GWAS of cerebellar volume.

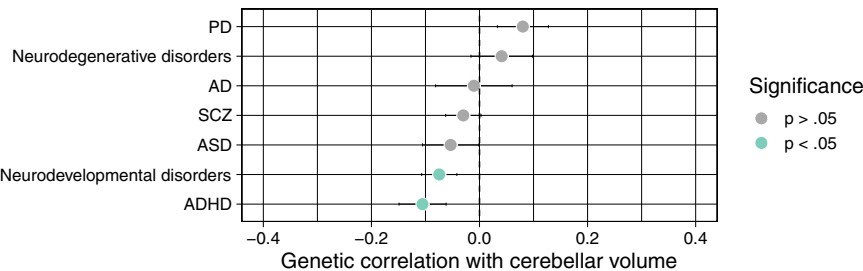

**Fig. 3 Global genetic correlations between cerebellar volume and disease.** Global genetic correlations and standard errors as estimated in LDSC between cerebellar volume and neurodevelopmental and neurodegenerative disorders (Supplementary Data 13). Although none of the traits survived Bonferroni correction for the number of traits tested, we did observe a slight gradient of negative $r_g$ with neurodevelopmental disorders to less negative and slightly positive $r_g$ with neurodegenerative disorders. To investigate this further, we meta-analysed the ADHD, ASD and SCZ summary statistics to represent the genetic signal of the overarching neurodevelopmental disorder dimension and similarly meta-analysed PD and AD summary statistics to capture the genetic signal of the neurodegenerative disorder dimension (see "Methods").

in both the PD and cerebellar volume GWAS. In AD and the neurodegenerative disorders meta-analysis phenotype, a positive correlation with cerebellar volume could be observed in a locus on chromosome 16 (Table 1). The lead SNP of this GWS AD locus is an intronic variant in *KAT8* and multiple significant eQTL variants in this locus influence *KAT8* expression in brain tissue including the cerebellum[38]. *KAT8* encodes for lysine acetyltransferase 8 that acetylates histone H4 at lysine 16 (H4K16ac)[39]. Alzheimer's disease patients show large amounts of H4K16ac loss compared to normal aging, especially close to Alzheimer's disease GWAS hits[40]. *KAT8* also seems to be vital for Purkinje cell survival[41]. For schizophrenia, we observed Bonferroni-corrected significant local $r_g$ with cerebellar volume in a positive direction on chromosome 12 and a negative direction on chromosome 19. The same loci were also Bonferroni significantly correlated between cerebellar volume and the meta-analysed neurodevelopmental disorders. The locus on chromosome 12 was GWS in both our GWAS as in the SCZ GWAS and includes 21 positionally mapped cerebellar-volume genes and 6 cerebellar-volume genes that were mapped through eQTL associations. The top hit for SCZ in this locus (rs2102949) was an intronic variant in *MPHOSPH9* that was also an eQTL in cerebellar tissue for *SETD8*, *CCDC62*, *PITPNM2* and *RP11-282O18.3*. *MPHOSPH9* encodes a phosphoprotein highly

expressed in the cerebellum, but its function is not well understood[42]. The negatively correlated locus on chromosome 19 reached GWS in SCZ, but not in cerebellar volume. Within this locus, the most significant SCZ signal came from variants located in the *GATAD2A* gene which is considered one of the promising causal genes for schizophrenia[43]. GATAD2A is part of a protein complex named nucleosome remodelling and deacetylase (NuRD), which is an important epigenetic regulator in granule neuron synapse formation and connectivity during sensitive time windows of cerebellar development[44].

In addition to computing $r_g$ within these loci, we ran colocalization analyses to determine whether cerebellar volume shared the same or a different causal variant as the neurodevelopmental and neurodegenerative disorders in these loci (Supplementary Data 15). Given that the posterior probabilities for neither hypothesis 3 (traits have different causal variants) nor hypothesis 4 (traits share the same causal variant) reached a convincing threshold (80%), we cannot conclude how the genetic signals of cerebellar volume and the neurodevelopmental and neurodegenerative disorders within these loci relate on a single-variant level.

**Polygenic score prediction and lead SNP validation.** We determined the robustness of our GWAS findings with out-of-

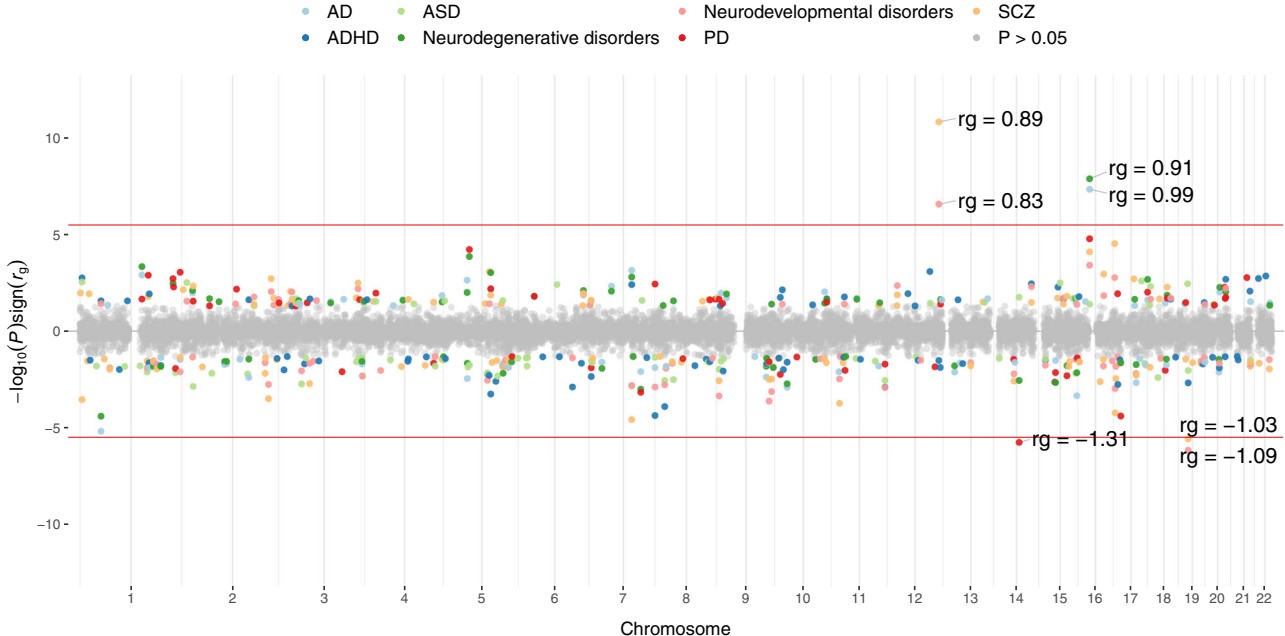

**Fig. 4 Local genetic correlations between cerebellar volume and disease.** Local genetic correlations ($r_g$) between cerebellar volume and neurodevelopmental and neurodegenerative disorders. The red line indicates the significance threshold, Bonferroni corrected for the number of loci tested in all seven traits. As during global $r_g$, we meta-analysed the ADHD, ASD and SCZ summary statistics to represent the genetic signal of the overarching neurodevelopmental disorder dimension and similarly meta-analysed PD and AD summary statistics to capture the genetic signal of the neurodegenerative disorder dimension (see "Methods").

**Table 1 Local genetic correlations between cerebellar volume and disease.**

| Trait | Locus | | | Local $r_g$ | $P$ value | Local $h^2_{shared}$ |
|---|---|---|---|---|---|---|
| | Chromosome | Start | End | | | |
| PD | 14 | 67,991,028 | 69,355,606 | −1.31 | $1.83 \times 10^{-6}$ | $-3.71 \times 10^{-4}$ |
| AD | 16 | 29,694,822 | 31,380,596 | 0.98 | $4.73 \times 10^{-8}$ | $3.72 \times 10^{-4}$ |
| NDG | 16 | 29,694,822 | 31,380,596 | 0.91 | $1.36 \times 10^{-8}$ | $4.02 \times 10^{-4}$ |
| SCZ | 12 | 121,721,831 | 124,710,880 | 0.89 | $1.58 \times 10^{-11}$ | $2.43 \times 10^{-3}$ |
| | 19 | 18,506,815 | 19,873,269 | −1.03 | $2.77 \times 10^{-6}$ | $-6.81 \times 10^{-4}$ |
| NDV | 12 | 121,721,831 | 124,710,880 | 0.83 | $2.78 \times 10^{-7}$ | $1.29 \times 10^{-3}$ |
| | 19 | 18,506,815 | 19,873,269 | −1.09 | $7.16 \times 10^{-7}$ | $-6.89 \times 10^{-4}$ |

Local genetic correlations ($r_g$) (Bonferroni-corrected $\alpha = 3.18 \times 10^{-6}$) between cerebellar volume and PD = Parkinson's disease, AD = Alzheimer's disease, SCZ = schizophrenia, NDV = meta-analysis of neurodevelopmental disorders (ASD, ADHD, SCZ), NDG = meta-analysis of neurodegenerative disorders (AD,PD). Local shared SNP-heritability was calculated as $h^2_{shared} = local r_g * \sqrt{local\, h^2_{trait\,1} * local\, h^2_{trait\,2}}$.

sample polygenic score (PGS) prediction using both a classical $P$ value thresholding and clumping (PRSice-2[45]) as well as a Bayesian (LDpred2[46]) method. Hyperparameter(s) were first optimised in a target set ($N = 1971$) before the best model was applied in a validation set ($N = 1983$). As shown previously[47], LDpred2-based PGS explained more variance in cerebellar volume (4.53%) in our validation set than PRSice-2-based PGS (2.36%; Supplementary Data 16). It is known that the prediction by PGS can be substantially lower than the $h^2_{SNP}$ when GWAS sample sizes are low[48], which is often the case in imaging-based GWAS. The variance explained by PGS will eventually climb close to $h^2_{SNP}$ estimates if GWAS sample sizes increase, as effect sizes are then estimated with less error and differences between base and target samples will decrease[48].

In addition, we were able to replicate 20 of our 37 discovery lead SNPs in our replication sample at $P < 0.05$ and 1 of 37 at $P < 5 \times 10^{-8}$ (Supplementary Data 17). The replication sample showed polygenic signal ($\lambda_{GC} = 1.029$) and was not confounded (LDSC$_{intercept} = 1.009$, SE $= 0.007$). Given the winner's curse

corrected discovery associations and the sample size in the discovery and the replication phase (see "Methods"), we expected to replicate 13 (exact 13.12) and 1 (1.17) SNPs at $\alpha = 0.05$ and $\alpha = 5 \times 10^{-8}$, respectively. The observed replication numbers corresponding and higher to the expected level emphasise the presence of a replicable genetic signal, especially at a sub-GWS level.

## Discussion

This study was designed to gain more insight into the genetic architecture of cerebellar volume. Our GWAS results in a substantial SNP-based heritability (39.82%) of cerebellar volume, with enrichments in super-enhancers and active promoters. The 30 GWS loci include SNPs that influence cerebellar gene expression levels or affect amino acid sequence. In total, 85 GWS genes were found to be associated with cerebellar volume, but not specific for a cerebellar cell type, developmental stage or biological pathway. Lastly, we identify specific loci that share genetic signals between Alzheimer's disease and Parkinson's disease and

cerebellar volume and between schizophrenia and cerebellar volume. In this study, we stratified cerebellar volume heritability, looked for convergence of the genetic signal, fine-mapped GWS cerebellar volume loci, and locally correlated the genetics of cerebellar volume and the neurodevelopmental and neurodegenerative disorders it is involved in.

Multiple results emphasise the importance of neurodevelopment in the genetics of cerebellar volume. First, both the enrichments of cerebellar volume SNPs, as well as heritability in regions with H3K4me1, H3K27ac and H3K9ac and chromatin states six and seven (H3K4me1), stress the importance of gene regulatory enhancer elements. An animal study[16] observed a strong overlap between regions with H3K4me1 and H3K27ac and regions with highly dynamic chromatin accessibility in cerebellar granule precursors and postmitotic cerebellar granule neurons. These dynamic regions enhance developmentally regulated increases in cerebellar granule neuron gene expression necessary for neuronal differentiation and function[16]. Second, we observe that the association between cerebellar volume genes and pre-natal cerebellar gene expression is stronger than that of post-natal gene expression. Third, we highlight credible causal SNPs, exonic nonsynonymous SNPs and eQTLs in and for genes with functionalities in neurodevelopment, including *PAX3, SLC39A8, SBNO1, ZNF789, PAPPA, MSI1, ZNF462, WASHC3, PARPBP, IGF1* and *C1orf185*. Our top hit, *PAPPA*, is most strongly expressed in the placenta[49]. The last trimester of pregnancy is a critical period for cerebellum growth with a fivefold increase in volume[50]. By cleaving insulin-like growth factor binding protein (IGFBP)−4 from IGF1, PAPPA promotes the availability of IGF1 and increases the probability that IGF1 binds its receptor[49] and activates various intracellular signalling pathways (such as the PI3kinase-Akt pathway, promoting cell growth and maturation[51]). IGF1 receptors are most abundantly found in brain regions rich of neurons such as the olfactory bulb, dentate gyrus and cerebellar cortex[51,52]. The cerebellum hosts two-thirds of the brain's neurons[53]. Examinations in preterm rabbit pups have indeed suggested that the loss of placental support is associated with lower levels of IGF1, decreased cerebellar external granular layer proliferation and Purkinje cell maturation[50]. Cerebellar neurogenesis also continues after birth and is accompanied with increased post-natal IGF1 levels[54]. Infant group studies have noted a positive correlation between IGF1 levels and total brain volume, with cerebellar volume showing the strongest association[55]. In mice, cerebellar growth is suggested to result from increases in cell numbers[56], as overexpression of *IGF1* caused a volumetric increase of the internal granular and molecular layer together with a respective increased number of granules and Purkinje cells[56].

*KAT8* is another gene that influences the number of Purkinje and granular cells, as indicated by a study examining *mMof* (the mouse homologue of *KAT8*) deficient mice[41]. The current study reveals local genetic correlation between Alzheimer's disease and cerebellar volume in a locus that includes the Alzheimer's lead SNP in *KAT8* and multiple variants influencing *KAT8* expression in brain tissue including the cerebellum[38]. Although the study showing large amounts of H4K16ac loss in Alzheimer's disease was performed in the temporal cortex[40], the authors note that cerebellar tissue is available for the same sample, which would be interesting to use in future research given this local genetic correlation. Genetic correlations between schizophrenia and cerebellar volume are observed in two loci that include *MPHOSPH9* and *GATAD2A* as most significant signal in schizophrenia. GATAD2A is part of the chromatin remodelling NuRD complex, which binds genome-wide active enhancers and promotors in embryonic stem cells to enable access for transcription factors to influence gene expression[44]. Especially in the cerebellar cortex, depletion of NuRD leads to impaired development of granule

neuron parallel fibres and Purkinje cell synapses[57]. Interestingly, this locus is not only associated with schizophrenia, but also with an opposing direction of effect for bipolar disorder[58]. Note, however, that we could not conclude whether these specific variants are the shared causal variants for both the disorder as cerebellar volume and results should be interpreted with this in mind.

We also observed null results for global genetic correlations with individual disorders and for the enrichment of cerebellar volume genes in pathways, developmental stage, human accelerated genes or cell types. A possible explanation is that the substantial heritability of cerebellar volume is related to the genetic signal being divided across many genes, leading to a decreased likelihood that most of the genetic variance lies in the gene set of interest[9]. Also note that postmortem cerebellar tissue can be scarce, for example resulting in a different single donor per developmental stage in the Brainspan database. More, and more robust cerebellum-specific gene sets would therefore be interesting for future studies. Another possibility is that we did not have sufficient power to detect significant results, because of our relatively small sample size. The last decade of GWAS has proven sample size to be crucial for discovering the often small genetic effects[59]. This underscores the need for larger sample sizes for future neuroimaging-genetics studies, though this is a time-consuming and costly challenge.

An important topic that needs to be addressed in neuroimaging-genetics studies is specificity. Structural properties (such as surface area, thickness or volume) are highly correlated across brain regions, which makes finding genetic variants that go beyond global brain effects and are potentially specific to brain regions challenging. In our study, we included total brain volume as a covariate in our GWAS to differentiate our findings from results on total brain volume. We are aware that such corrections do not guarantee pure specificity and our results reflect processes that are not exclusively involved in cerebellar volume (e.g. *IGF1* is expressed in various tissue types and has widespread functions). However, the fact that our results can be interpreted as concordant with previous literature from the developing cerebellum or cerebellar cell types, strengthens our conclusion that these processes are involved in cerebellar volume.

A number of limitations of our study must be considered. Our sample predominantly consists of British participants of European ancestry. The effect of the European bias in the majority of GWAS remains relatively unknown, as the extent to which GWAS results can be transferred to other populations depends on many factors[60]. However, the call for a multi-ancestry GWAS approach in future studies has been gaining increasing support since it will lead to a benefit of science to all populations. Two other characteristics of our sample are the relatively high age and socioeconomic status of subjects. Genetic correlations can be influenced by genetic overlap with socioeconomic status traits, as was demonstrated for mental traits previously[61]. We, therefore, corrected for age effects and Townsend deprivation index (TDI; a proxy of socioeconomic status) within our sample, to reduce this bias in our genetic correlation analyses between cerebellar volume and neurodevelopmental and neurodegenerative disorders. Nevertheless, these sample characteristics highlight the need for other, more diverse, large neuroimaging-genetics datasets, which additionally could also aid well-powered independent replication of discovery GWAS findings. With the present data at hand, we were able to internally validate genetic variants from our discovery sample in a holdout sample to understand the generalisability of our findings.

In sum, we conducted the most comprehensive study of the genetics underlying cerebellar volume to date. The genetic signal for cerebellar volume measured in middle age shows a strong

neurodevelopmental character. The variants and genes identified have previously been shown to influence the development of the cerebellum via chromatin accessibility of genomic elements that influence gene expression, cell survival/death and cell growth stimulation. These insights underscore the importance of perinatal neurodevelopment for the size of a key brain structure later in life that is essential for cognitive functioning and affected in disorders.

## Methods

**Sample**. A flowchart that describes all Methods used in this manuscript is displayed in Supplementary Fig. 4.

All data used in this study originate from volunteer participants of the UK Biobank who provided written informed consent. The study was conducted under application number 16406. The UK Biobank obtained ethical approval from the National Research Ethics Service Committee North West–Haydock (reference 11/NW/0382) and provides researchers worldwide with a data-rich resource, including SNP-genotypes and neuroimaging data. SNP-genotypes were released for $N = 488,377$ participants in March 2018, with neuroimaging data available for a subsample of N = 40,682 individuals[62]. From the ~20,000 subjects released in January 2020, we drew a random list of 5000 subjects for replication purposes. From the 40,682 discovery and replication subjects available, $N = 9$ were excluded due to later withdrawn consent and 6058 individuals were excluded by us because of UKB-provided relatedness (subjects with the most inferred relatives, third-degree or closer, were removed until no related subjects were present), discordant sex, or sex aneuploidy. Additionally, individuals of European descent were included if their projected principal component score was closest to and <6SD (based on Mahalanobis distance) from the average principal component score of the European 1000 Genomes sample ($N = 2187$ non-European exclusions), as has been described in previous publications by our group[63]. Phenotype data were quality-controlled as described below ($N = 513$ missing phenotypes and $N = 121$ outliers) and was matched with the quality-controlled genotype data: we continue by reporting the maximum per-SNP sample size ($N = 31,392$). After splitting subjects based on the randomly drawn replication list, we arrived at two samples: the discovery sample ($N = 27,486$) aged $M = 63.55$ (SD = 7.52) years with 52.50% females and the replication sample ($N = 3906$) aged $M = 64.91$ (SD = 7.30) years with 54.58% females. Supplementary Data 18 gives an overview of sample quality and exclusion criteria.

**SNP genotype data**. All participants of which data were used in this study were genotyped on the UK Biobank™ Axiom array by Affymetrix, covering 825,927 single-nucleotide polymorphisms (SNPs). Both quality control and imputation were executed by the UK Biobank, using the combined Haplotype Reference Consortium and the UK10K haplotype panel as a reference and resulting in 92,693,895 SNPs. Imputed variants were converted to hard-call SNPs using a certainty threshold of 0.9.

Prior to downstream analyses, we performed our own additional quality control procedure. SNPs with a low imputation score (INFO < 0.9), low minor allele frequency (MAF < 0.005) and high missingness (>0.05) were excluded as well as multiallelic SNPs, indels, and SNPs without unique rs identifiers. This resulted in a total of 9,380,668 SNPs.

**Neuroimaging data**. T1-weighted and T2-weighted images formed the basis of the cerebellar volume estimates used in this study. The UK Biobank scanning protocol and processing pipeline is described in the UK Biobank Brain Imaging Documentation[64]. The processed and quality-controlled[65] estimates of cerebellar white and grey matter per hemisphere were downloaded from the UK Biobank and merged to represent an estimate of total cerebellar volume. In total, $n = 513$ individuals had missing cerebellar volume estimates and $n = 121$ outliers were removed as having a cerebellar volume >3 median absolute deviations (MAD). Mean cerebellar volume in the discovery sample was 143,580 mm$^3$ (SD = 14,231 mm$^3$) and 143,260 mm$^3$ (SD = 14,095 mm$^3$) in the replication sample, following normal distributions.

**Genome-wide SNP-based association study**. Discovery and replication SNP-based genome-wide association studies (GWAS) were performed in PLINK version 2.00[66] to identify and replicate common genetic variants that contribute to inter-individual cerebellar volume variability. Principal component analysis (PCA) was applied on both unrelated European neuroimaging samples using FlashPCA2[67] to correct for population stratification. We selected a stringent set of independent ($r^2 < 0.1$), common (MAF > 0.01), and genotyped SNPs or SNPs with very high imputation quality (INFO = 1) as features in PCA ($n = 145,432$). The first 20 genetic principal components (PCs) served as covariates together with sex, age, genotype array, Townsend deprivation index (TDI; a proxy of socioeconomic status), and several neuroimaging-related confounders that were recommended by Alfaro-Almagro and colleagues[68]. These included handedness, scanning site, the use of T2 FLAIR in Freesurfer processing, intensity scaling of T1, intensity scaling

of T2 FLAIR, scanner lateral (X), transverse (Y) and longitudinal (Z) brain position, Z-coordinate of the coil within the scanner, and total brain volume. The latter is important since cerebellar volume evidently relates to the volume of the total brain, but we are interested in the genetic signal specific to the cerebellum. A linear regression model with additive allelic effects was fitted for each SNP to detect potential genetic effects on cerebellar volume. Variants on XY chromosomes were treated like autosomal variants with male X genotypes counted as 0/1 dosage. The alpha level for SNPs reaching genome-wide significance was adjusted from $\alpha = 0.05$ to $\alpha = (0.05/1,000,000 =) \ 5 \times 10^{-8}$ according to the Bonferroni correction for multiple testing. Unstandardised beta values (B) were standardised ($\beta$) using the following equation:

$$\beta = \frac{Z\ score}{\sqrt{(2 * MAF * (1 - MAF)) * (N_{discovery} + Z\ score^2)}} \ with\ Z\ score = \frac{B}{SE} \quad (1)$$

**(Partitioned) SNP heritability**. Linkage disequilibrium score (LDSC) regression was used to estimate how much of cerebellar volume variability could be explained by additive common genetic variation[69]. This so-called SNP heritability, or $h^2_{SNP}$, captures only the proportion of additive genetic variance due to LD between the assayed and imputed SNPs and the unknown causal variants. Precomputed LD scores based on the 1000 Genomes European data were used for this purpose.

Stratified LDSC regression was performed per functional genetic category[70] to investigate if certain sites in the human genome contribute disproportionately to $h^2_{SNP}$ estimates. Enrichment of $h^2_{SNP}$ in one of the 28 categories was calculated as the proportion of $h^2_{SNP}$ divided by the proportion of SNPs. The alpha level for significance was adjusted from $\alpha = 0.05$ to $\alpha = (0.05/28 =) \ 1.79 \times 10^{-3}$ according to the Bonferroni correction for multiple testing.

**Functional annotation and mapping of cerebellar volume-associated SNPs**. The discovery summary statistics from the SNP-based GWAS served as input for the web-based platform FUMA[18] to functionally map and annotate genetic associations with cerebellar volume. In order to do so, FUMA first determined which genome-wide significant SNPs were independent from one another ($r^2 < 0.6$). SNPs in linkage disequilibrium (LD) with independent significant SNPs ($r^2 \geq 0.6$) were defined as candidate SNPs (using both summary statistics and 1000 G Phase 3 EUR[71]). Second, a more stringent cut-off of $r^2 < 0.1$ was used to determine which independent significant SNPs could be defined as lead SNPs. Third, genomic loci were represented by the lead SNP with the lowest $P$ value in the locus. All independent significant SNPs $r^2 < 0.1$ with LD blocks within 250 kb distance and independent significant SNPs $r^2 \geq 0.1$ were assigned to the same genomic risk locus. Fourth, all candidate SNPs were annotated using ANNOVAR (1000 G Phase 3 EUR as reference panel[71]), RegulomeDB score[24] and ChromHMM[19] (using the common chromatin states in the available adult brain samples). Enrichment of candidate SNPs falling into the categories of these annotations were calculated with Fisher's Exact Test and $P$ values adjusted for multiple testing conformed to Bonferroni correction. Fifth, annotated SNPs were mapped to one or multiple genes by two different strategies. Positional mapping was based on physical distances (<10-kb window). Expression quantitative trait locus (eQTL) mapping was based on established associations between SNPs and a gene's (in cis, <1 Mb window) expression profile in the cerebellum and cerebellar hemisphere from Genotype-Tissue Expression (GTEx)[72] v8 and cerebellar cortex from BRAINEAC[73] databases.

**Statistical fine-mapping of cerebellar volume-associated loci**. Focussing on the most significant independent SNP per locus (lead SNP, described above) is not always preferable, since the most significant variants are not necessarily causal variants[74]. Therefore, statistical fine-mapping methods have been developed to determine the probability of variants being causal. Here we applied FINEMAP[26] to genomic loci as defined in FUMA. FINEMAP is a Bayesian statistical fine-mapping tool that estimates the posterior probability of a specific model, by combining the prior probability and the likelihood of the observed summary statistics. The posterior probabilities can in turn be used to calculate the posterior inclusion probability (PIP) of a SNP in a model and the minimum set of SNPs needed to capture the SNPs that most likely cause the association[74]. We set the maximum number of causal SNPs to 10. We additionally used LDstore[75] to estimate the pairwise LD matrix of SNPs from quality-controlled genomic data of the discovery sample. 95% credible set SNPs were defined by first summing the posterior probabilities of each model (starting from the highest probability model) until we reached a total of 95% and secondly taking all unique SNPs existing in those models. Only those SNPs with a posterior inclusion probability (PIP) > 0.95 were used for interpretation.

**Gene-based GWAS**. SNP-based GWAS approaches can suffer from power-related issues, so we combined the information from neighbouring variants within a single gene to boost power. A multi SNP-wise model for gene-based GWAS was used in MAGMA (Multi-marker Analysis of GenoMic Annotation)[76] v1.08. The discovery of SNP-based GWAS summary statistics served as input for the gene-based GWAS and covered 18,852 genes, the UKB European population was used as an ancestry reference group. The alpha level for genes reaching genome-wide significance was

adjusted from $\alpha = 0.05$ to $\alpha = (0.05/18,852 =)\ 2.65 \times 10^{-6}$ according to Bonferroni correction for multiple testing.

**Functional gene-set analysis.** The summary statistics resulting from gene-based GWAS were further investigated using a pathway-based association analysis in MAGMA to test for enrichment of cerebellar volume-associated genes in established biological annotated pathways. We used gene sets from Gene Ontology (GO) molecular functions, cellular components and biological processes, and curated gene sets from MsigDB v7.0 (sets C2 and C5)[31]. Protein-coding genes served as background genes. The alpha level for gene sets reaching significance was adjusted from $\alpha = 0.05$ to $\alpha = (0.05/7,246 =)\ 6.90 \times 10^{-6}$ according to the Bonferroni correction for multiple testing.

**Temporal gene-set analysis.** In order to identify a developmental window important for cerebellar volume, we tested the relationship between age-specific gene expression profiles and cerebellar volume-gene associations (gene-based summary statistics). For this purpose, age-specific gene expression profiles were downloaded in the form of RNA sequencing data of the BrainSpan Atlas of the Developing Human Brain[29]. We extracted gene-level RPKM values of 29 cerebellar cortex samples. Of 52,376 available genes, 29,114 genes were filtered out because RPKM values were not >1 in at least one age stage, 7828 genes were excluded because of a missing EntrezID. Finally, we included genes on the SynGO[77,78] list of brain genes resulting in a set of 14,172 genes. RPKM was winsorized at 50, after which we log-transformed RPKM values with pseudocount 1. Eight samples were of the same age (21 postconceptional weeks (pcw), 4 months, 3 years, 8 years), therefore $\log 2(\text{RPKM} + 1)$ values were averaged across samples of the same age. The $\log 2(\text{RPKM} + 1)$ value per gene was averaged across ages to use as a covariate. A one-sided conditional MAGMA gene-property analysis was performed, testing the positive relationship between age specificity and the genetic association of genes (gene-based summary statistics).

To determine whether there was a general difference between the effects of pre- and post-natal gene expression, we tested the difference in mean effect between those two groups of timepoints as follows. From the MAGMA output, we can model the distribution of the marginal effect estimate $\hat{\beta}_j$ of each timepoint $j$ as

$$\hat{\beta}_j \sim \mathrm{N}\left(\beta_j, S_j^2\right) \tag{2}$$

with $\beta_j$ the unknown true effect and $S_j$ the standard error. As such, for the vector $\hat{\beta}$ of all $K$ estimates jointly we obtain the distribution $\hat{\beta} \sim \mathrm{MVN}(\beta, SRS)$. Here, $S$ is a diagonal matrix containing the standard errors of all the estimates, and $R$ is the sampling correlation matrix which we can approximate by the correlation matrix of the gene expression timepoints. We now define $\mu_{\text{pre}}$ as the mean of the $\beta_j$ of prenatal timepoints, such that

$$\mu_{\text{pre}} = \frac{1}{K_{\text{pre}}} \sum_{j \in \text{pre}} \beta_j \tag{3}$$

with $K_{\text{pre}}$ the number of pre-natal timepoints, and likewise $\mu_{\text{post}}$ as the mean of the $\beta_j$ of the post-natal timepoints. These $\mu_{\text{pre}}$ and $\mu_{\text{post}}$ are estimated using the corresponding means of the $\hat{\beta}_j$. With this, we can now test the null hypothesis $H_0 : \mu_{\text{pre}} = \mu_{\text{post}}$ that the means of the effects of pre- and post-natal gene expression are the same. Under this null hypothesis, the difference in estimated means $\Delta_{\hat{\mu}} = \hat{\mu}_{\text{pre}} - \hat{\mu}_{\text{post}}$ is distributed as

$$\Delta_{\hat{\mu}} \sim \mathrm{N}\left(0, D^T SRSD\right) \tag{4}$$

which can be used to compute a $P$ value. In this expression, $D$ is a length $K$ coding vector with a value for each timepoint, being equal to $\frac{1}{K_{\text{pre}}}$ for pre-natal timepoints and $\frac{-1}{K_{\text{post}}}$ for post-natal timepoints.

**Gene-set analysis in cerebellar cell types.** Understanding the cell type-specific expression of cerebellar volume genes may help understand the circuitry basis of cerebellar volume. We used the cell type-specificity analysis as implemented in FUMA to test whether genetic variants for cerebellar volume converge on a specific murine cerebellar cell type identified in the DropViz Level 1 database[33]. This MAGMA gene-property analysis uses gene expression values in specific cell types as gene properties and aims to test the relationship between this property and cerebellar volume-gene associations. Several technical factors, such as gene length and correlations between genes based on LD, are included to control for confounding effects[79]. The alpha level for cerebellar cell types reaching significance was adjusted from $\alpha = 0.05$ to $\alpha = (0.05/9 =)\ 5.56 \times 10^{-3}$ according to the Bonferroni correction for multiple testing.

**Evolutionary gene-set analysis.** The cerebellum underwent a significantly faster size increase throughout evolution than predicted by the change in neocortex size[80]. We therefore investigated if any evolutionary signatures could be observed in our GWAS. For this purpose, a gene set was curated from genes located in human accelerated regions (HARs) and used during MAGMA's gene-set analysis. These previously identified regions are highly different between humans and other species and are suggested to have potential roles in the evolution of human-specific

traits[32]. One thousand five hundred seventy-one of the HAR-associated genes were present in the cerebellar volume gene-based GWAS summary statistics. The alpha level for the HAR gene-set reaching significance was $\alpha = 0.05$.

**Global and local genetic correlations with neurodevelopment and neurodegeneration.** A statistically significant genetic correlation ($r_g$) between two traits reflects the existence of a shared genetic profile and is based on correlations in genome-wide effect sizes across traits. Cross-trait LDSC obviates the need of measuring both traits per individual but allows for full sample overlap at the same time[81]. Here, cross-trait LDSC regression was used to compute $r_g$ between cerebellar volume discovery SNP-based GWAS summary statistics and five psychiatric and neurological disorders for which well-powered ($N > 20,000$) GWAS summary statistics were publicly available. These included the neurodevelopmental disorders autism spectrum disorder (ASD)[82], attention deficit hyperactivity disorder (ADHD)[83] and schizophrenia (SCZ)[84] and the neurodegenerative disorders Parkinson's disease (PD)[85] and Alzheimer's disease (AD)[38].

Where the global $r_g$ is an average correlation of genetic effects across the genome, local $r_g$ can identify loci that show genetic similarity between traits. Local $r_g$ were estimated in this study using SUPERGNOVA[86], using the same cerebellar volume and neurodevelopmental and neurodegenerative disorder summary statistics as input. Note that, prior to local $r_g$ estimation, SUPERGNOVA does not filter loci on univariate association signal and that local correlation estimates lower than $-1$ or higher than 1 may occur in loci with numerically unstable estimates of heritability[86]. The alpha level for a local $r_g$ reaching significance was adjusted for the number of loci tested across all traits ($\alpha = 3.18 \times 10^{-6}$).

After observing the results, we decided to meta-analyse the neurodegenerative disorders and meta-analyse the neurodevelopmental disorders to reduce the error term. Variants with a minor allele count (MAC) < 100 or that were not present in both (neurodegenerative disorders) or two out of three (neurodevelopmental disorders) GWAS summary statistics were excluded from the analysis. We used the software tool mvGWAMA[38] to perform two effective sample size

$$N_{eff} = \frac{4}{\frac{1}{N_{cas}} + \frac{1}{N_{con}}} \tag{5}$$

Weighted analyses based on $P$ values. mvGWAMA uses the bivariate LDSC intercept to correct for sample overlap that potentially exists between the samples underlying the input GWAS summary statistics. Sample overlap between pairs of GWAS was estimated by

$$N_{overlap} = i_{bivariate} \times \sqrt{N_1 \times N_2} \tag{6}$$

As described earlier in ref. [87], resulting in the following: ADHD/SCZ (4,457), ASD/SCZ (2,048), ADHD/ASD (18,041), AD/PD (5438). In mvGWAMA, we performed a two-sided meta-analysis, meaning that the direction was aligned and conversion between $P$ values and z-scores was two-sided. Using the output meta-analysis summary statistics, global and local genetic correlations were estimated as described above. The alpha level for global and local genetic correlations was additionally adjusted for the number of traits tested according to the Bonferroni correction for multiple testing. For all global and local genetic correlations, the shared heritability between cerebellar volume and the other trait was calculated as

$$h_{shared}^2 = r_g * \sqrt{h_{trait\,1}^2 * h_{trait\,2}^2}. \tag{7}$$

**Colocalization of significant $r_g$ loci.** After obtaining significant genetically correlated loci as described above, we investigated if the summary statistics support a shared causal variant for both traits. We ran colocalization analyses with the *coloc* R package[88] for the genetically correlated loci that reached significance. Coloc is a Bayesian method that provides posterior probabilities for five hypotheses (H$_0$ = no association with either trait; H$_1$ = association with trait 1, not with trait 2; H$_2$ = association with trait 2, not with trait 1; H$_3$ = association with trait 1 and trait 2, two independent SNPs; H$_4$ = association with trait 1 and trait 2, one shared SNP). We called the signals colocalized when the posterior probability of H4 > 80%.

**Genetic architecture comparison with cerebral and subcortical volume.** In order to place our cerebellar volume GWAS into perspective, we compared its genetic architecture with the other two major volumes of the brain: cerebral volume and subcortical volume. Two SNP-based GWAS were performed for total subcortical volume (thalamus, caudate, putamen, pallidum, hippocampus, amygdala, accumbens, ventral diencephalon, and substancia nigra) and total grey + white cerebral volume, using the same pipeline and covariates as for our cerebellar volume discovery GWAS described above. First, we used univariate MiXeR[17] to describe the number of genetic variants contributing to each trait, their polygenicity ($\pi$: the proportion of independent causal SNPs) and discoverability ($\sigma\beta$: the variance of the effect sizes of causal SNPs). $\pi$ ranges between 0 and 1 and higher $\pi$ values indicate higher polygenicity. High $\sigma_\beta^2$ indicates a high level of discoverability of causal SNPs for the trait. 1000 Genomes EUR was used as a reference panel and the MHC region (chr6: 26–34 Mb) was excluded from all three SNP-

based GWAS summary statistics. To complete our overview of the genetic architecture per trait, we plotted the minor allele frequencies of (FUMA-defined) independent significant SNPs against their squared standardised effect sizes. Second, we used LDSC and SUPERGNOVA to compute global and local $r_g$ between cerebellar, cerebral and subcortical volume. The alpha level for significance was Bonferroni corrected for the number of traits (global: $\alpha = 0.05/3 = 1.66 \times 10^{-2}$) and loci investigated (local: $\alpha = ((0.05/2257 \text{ loci})/3 =) \ 7.38 \times 10^{-6}$).

**Replication of discovery lead SNPs**. The replicability of our GWAS results were evaluated by assessing the associations of 37 discovery lead SNPs in our replication GWAS. For this purpose, we compared observed numbers of replicated lead SNPs with numbers expected under a natural alternative hypothesis introduced in ref. [89] (Supplementary Information 1.8). Under this hypothesis, the probability that discovery lead SNP $i$ is significant in the replication sample is

$$P(sig_i) = \Phi\left(-\frac{|\beta_i|}{\sigma_{rep,i}} + \Phi^{-1}\left(\frac{\alpha}{2}\right)\right) + \left[1 - \Phi\left(-\frac{|\beta_i|}{\sigma_{rep,i}} - \Phi^{-1}\left(\frac{\alpha}{2}\right)\right)\right] \quad (8)$$

with $\alpha$ representing an alpha level of 0.05 and $5 \times 10^{-8}$, $\Phi$ the cumulative normal distribution function, $\Phi^{-1}$ the inverse normal distribution function, $\sigma_{rep,i}$ the standard error of SNP $i$ in the replication GWAS and $\beta_i$ the winner's curse adjusted association estimate of SNP $i$. Winner's curse is the occurrence of overestimated effect sizes that are induced by significance thresholding[90]. Correction for winner's curse was performed using the Conditional Likelihood method[91] in the winnerscurse R package, which uses both the discovery and replication betas and standard errors for correction. Finally, the number of SNPs that is expected to show significance was defined as

$$\sum_i P(sig_i). \quad (9)$$

**Polygenic scores**. In order to estimate how much variance in cerebellar volume could be explained by our GWAS findings, we calculated polygenic scores using the discovery summary statistics (base set) in PRSice-2[45] and LDpred[46]. For this purpose, we randomly split the replication sample into a target set ($N = 1971$) and a validation set ($N = 1983$). A three-phase procedure was applied, with phase 1 representing the estimation of effect sizes in our discovery GWAS. In phase 2, SNP data (MAF > 0.1, chromosome X excluded) of the target sample was used to optimise $P$ value thresholding (PRSice-2) and hyper-parameters (LDpred2) to find the best fit model. We constricted our LDpred2 analyses to HapMap3 variants as recommended. All models were corrected for the same covariates as during GWAS analysis described above. During phase 3, the best model was fit on clumped SNP data of the validation set to be able to report the phenotypic variance explained and $P$-value unaffected by overfitting.

**Reporting summary**. Further information on research design is available in the Nature Research Reporting Summary linked to this article.

## Data availability

Genome-wide summary statistics are publicly available via https://ctg.cncr.nl/software/summary_statistics/ and GWAS Catalog. The individual-level data that support the findings of this study are available from UK Biobank but restrictions apply to the availability of these data, which were used under license no. 16406 for the current study, and so are not publicly available. Data are however available from the authors upon reasonable request and with permission of UK Biobank. All other data used in this study are publicly available via BrainSpan http://www.brainspan.org/static/download, SynGO https://www.syngoportal.org, DropViz http://dropviz.org/, MSigDB https://www.gsea-msigdb.org/gsea/msigdb/collections.jsp, Psychiatric Genomics Consortium http://www.med.unc.edu/pgc/results-and-downloads. Supplementary Data 3 contains source data underlying Fig. 1c. Supplementary Data 1 contains source data underlying Fig. 1d. Supplementary Data 13 contains source data underlying Fig. 3.

## Code availability

No new software was developed for this project, existing software and code are publicly available: FUMA http://fuma.ctglab.nl/, MAGMA https://ctg.cncr.nl/software/magma, mvGWAMA https://github.com/Kyoko-wtnb/mvGWAMA, LDSC https://github.com/bulik/ldsc, SUPERGNOVA https://github.com/qlu-lab/SUPERGNOVA, COLOC https://chr1swallace.github.io/coloc, FINEMAP http://www.christianbenner.com, MiXeR https://github.com/precimed/mixer, PRSice-2 https://www.prsice.info, LDpred2 https://privefl.github.io/bigsnpr/articles/LDpred2.html, PLINK https://www.cog-genomics.org/plink/, FLASHPCA https://github.com/gabraham/flashpca, winnerscurse R package https://amandaforde.github.io/winnerscurse/.

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

## Acknowledgements

D.P. was funded by The Netherlands Organization for Scientific Research (NWO VICI 453-14-005), NWO Gravitation: BRAINSCAPES: A Roadmap from Neurogenetics to Neurobiology (Grant No. 024.004.012), and a European Research Council advanced grant (Grant No, ERC-2018-AdG GWAS2FUNC 834057). The work of S.L. was supported by ZonMw Open Competition, project REMOVE 09120011910032. D.W. was funded by NWO Gravitation: BRAINSCAPES: A Roadmap from Neurogenetics to Neurobiology (grant no. 024.004.012). C.L. is funded by Hoffman-La Roche. The work of M.H. was supported by a VIDI (452-16-015) grant from the Netherlands Organization

for Scientific Research (NWO) and an ERC Consolidator of the European Research Council (101001062). The research has been conducted using the UK Biobank Resource (application no. 16406). Analyses were carried out on the Genetic Cluster Computer hosted by the Dutch National computing and Networking Services SURFsara. We thank J.P. Beauchamp for his clarifying emails about the holistic replication method.

## Author contributions

E.T., M.P.v.d.H. and D.P. contributed to the conception and design of the study. J.E.S. performed preprocessing of genetic data. E.T. performed all analyses. C.A.d.L., D.P.W. and K.M.K. provided data and analysis scripts in three analyses. E.T., S.C.d.L., J.E.S., D.P.W., K.M.K., C.A.d.L., M.N., M.P.v.d.H. and D.P. contributed to the interpretation of the data. E.T. has drafted the work. E.T., M.N., M.P.v.d.H. and D.P. substantively revised it and J.E.S., S.C.d.L., D.W., C.A.d.L. and K.M.K. revised it.

## Competing interests

The authors declare no competing interests.
