## [Peer Review File · Communications Biology]

Reviewers' comments:

Reviewer #1 (Remarks to the Author):

The aim of this paper is to investigate the convergence of cerebellar volume genetic associations in details. Compared with the existing literature, this paper focuses on a thorough interrogation of cerebellar volume at the level of genetic variants with detailed follow-up focussed on translating genetic loci into mechanistic hypotheses could elucidate why cerebellar volume varies between individuals and contributes to neurodevelopmental and neurodegenerative diseases.

- 1, Could the authors elaborate more on the similarity and difference between their results and those in Ref 8-10? For instance, are you GWAS results similar to those in the literature?
- 2, Could the authors elaborate more on how the replication sample was selected?

Reviewer #2 (Remarks to the Author):

Tissink and colleagues conduct a GWAS of cerebellar volume. Overall, the GWAS is well conducted and the analysis is technically sound. The paper is well written and easy to follow.

That said, my enthusiasm is dampened by the fact that I don't think the paper is telling anything new, and seems to be analyses run through a fairly standard pipeline. Given that summary statistics for this phenotype is available already, I am left wondering what this paper is adding to the field.

Would the authors consider additional lines of analyses to tell us something new about the cerebellum that's not quite fully known yet using the GWAS analyses? For instance, how does the genetic architecture compare to that of the cerebral volumes/SA? Are there specific evolutionary signatures that can be gleaned from the analyses of the GWAS data?

Reviewer #3 (Remarks to the Author):

Tissink and colleagues report on a genome-wide association study on MRI cerebellar volume in 31,392 individuals included in the UK Biobank. The trait is highly heritable which led them to find 29 genomic loci that pass genome-wide significance. The genome-wide summary statistics are well-behaved and through several downstream analyses they nominate candidate genes that affect cerebellar volume. Subsequently, they study enrichment of cell-types, pathways or developmental stages. Finally, they relate cerebellar volume to neuropsychiatric traits through estimating (local) genetic correlations.

To my knowledge this is the first GWAS on cerebellar volume (besides the preprint that is cited in this paper). The main analyses are solid and these data are therefore valuable to a wide scientific audience especially involved in genetics and neuroscience.

There are some aspects of this study that could be clarified.

- The methods describe MRI data is available for 40,682 individuals, but eventually 31,392 are included. This means that nearly a quarter of the data is lost due to quality control. Can the authors specify at which step how many samples failed?

- The methods state that 6,067 were excluded, but judging from the number more were lost (40,682 - 31,392 = 9,290), can the authors explain this discrepancy.

- The authors perform LDSC to show the GWAS was overall not confounded. Personally, I'm not sure if subsequently mentioning Lambda1000 - Lambda10,000 is that informative (as the N dimension is already taken into account in the LDSC h2 estimate and intercept). Furthermore the statement that Lambda increases with N is a characteristic of a polygenic trait in partially true, it is also a characteristic of a confounded GWAS that grows in size (LDSC is therefore more informative).

- I do not totally understand the purpose of the out of sample prediction analyses. Is it to illustrate robustness of the reported associations? Maybe it could be combined describing the replication efforts. That r2 for out of sample prediction is lower than h2 is expected from theory. If the authors want to present an optimal PRS other methods than clumping and thresholding might be better (Ni et al, Biol. Psych., 2021).

- Can the authors report the lambda and LDSC intercept for the replication GWAS?

- Did the authors perform a genome-wide meta-analysis of discovery and replication dataset (given the high number of replicated loci)?

- Which summary statistics are used for all downstream enrichment analyses (discovery or a meta-analysis of discovery + replication)?

- figure 1c: color scale of enrichment might be easier to read on log2() scale so it is symmetric.

- For the genetic correlations with neurodegenerative and neurodevelopmental traits, it might be good to state that there were no genetic correlations *after correction for multiple testing*.

- For the exonic variants: it could help to report the amino-acid change and effect (SD cerebellar volume) for all of them? Can they report SE or 95% CI for the effect size?

- line 173: not sure if MAGMA "confirms" as it is a result of an LD-informed aggregation test that includes the same SNP (and thus signal), "which is in line with" might be more appropriate.

- The title of paragraph "Convergence of cerebellar volume genes on pathways, cell types and developmental stage" suggests the authors did find enriched pathways, cell-types and developmental stages. Instead, the describe no strong evidence of convergence on cell-types, pathways or developmental stages (after correction for multiple testing). They could consider rephrasing the title of this paragraph more neutrally.

- The BRAINSPAN enrichment analysis is interesting. If I'm correct, the MAGMA analysis treats all timepoints as individual sets, and not the temporal ordering of these sets. The enrichment in pre-natal stages is not statistically significant (after correction for multiple testing), but the trend in p-values might lead the reader to speculate. Can the authors plot the regression coefficients + SE to see if there is a consistent trend of higher/lower expression in pre-natal stages or are these coefficients all over the place? Would binning in pre-natal, childhood, puberty, adolescence and adulthood samples increase power?

- The regional genetic correlation analyses highlights several loci that explain heritability in cerebellar volume and AD/PD/SCZ. An interesting question is whether this is due to two different causal variants in LD or is there an actual shared signal? This might be beyond the scope of this paper, but the authors could run colocalization analyses for this purpose. If there is indeed evidence for a shared causal signal (and not just LD), the description of gene functions in these regions are more relevant.

- The label on y-axis of fig 4 is a bit unclear, specifically the difference between ρ and r_g , maybe better to write P and r_g ? And clarify that labels in the plot correspond to r_g estimates.

- The authors rightfully acknowledge that it is hard to pinpoint the genetic correlations to cerebellar volume, as itself is correlated to the volume of other brain regions. Could multivariate Mendelian Randomisation studies help here? Of course, this could be beyond the scope of this study.

Thanks for sharing the summary statistics upon publication, will they be deposited in the GWAS catalog?

Please find below our point-by-point response to the comments raised by the reviewers together with the adjustments we made to the manuscript. The questions and comments of the reviewers are presented in *italics*, our response in normal typeset. We marked changes to the text in blue in the updated version of our manuscript.

Reviewer #1

The aim of this paper is to investigate the convergence of cerebellar volume genetic associations in details. Compared with the existing literature, this paper focuses on a thorough interrogation of cerebellar volume at the level of genetic variants with detailed follow-up focussed on translating genetic loci into mechanistic hypotheses could elucidate why cerebellar volume varies between individuals and contributes to neurodevelopmental and neurodegenerative diseases.

1. Could the authors elaborate more on the similarity and difference between their results and those in Ref 8-10? For instance, are you GWAS results similar to those in the literature?

We have followed up on this suggestion and have added a more detailed comparison between our results and those of refs 8-10 in Supplementary Note and Supplementary Table 19.

Supplementary Note, section “Comparison results with Smith *et al* and Zhao *et al*”, page 1, line 17

“Ref 8 (Smith *et al*) and Ref 9 (Zhao *et al*) have provided *en masse* summary statistics, including for the volume of cerebellar vermal lobules and lateral cerebellar grey and white matter (total cerebellar volume was not investigated). The reported h_{SNP}^2 estimated with GCTA-GREML varied largely across cerebellum subregions in the Zhao *et al* publication (M = 64.55%, SD = 10.64%), whereas lower and more stable LDSC-based h_{SNP}^2 estimates were observed (M = 27.95%, SD = 0.35%) by Smith *et al*. Smith *et al* do not provide any post-GWAS investigation, but Zhao *et al* report 19 non-overlapping loci, 69 mapped genes and no gene-sets to be significantly associated with at least one cerebellar volume subregion. We also observe 10 of those loci and 35 of those genes for total cerebellar volume, but additionally report 20 and 191 novel loci and genes respectively. This increase is likely due to our significantly larger sample size (N = 27,486) compared to Zhao *et al* (N = 19,629) and our cerebellum-specific gene-mapping protocol instead of using all brain-based eQTL and chromatin interaction reference data available with default settings by Zhao *et al*.”

Supplementary Note, section “Comparison results with Chambers *et al*”, page 1, line 31

“Ref 10 (Chambers *et al*) provides a GWAS of total cerebellar grey matter volume and reports 29 loci from GCTA-COJO (Table 1 of ref 10). Since we included a different phenotype (total cerebellar grey and white matter volume) and defined loci in FUMA, comparing loci one-on-one can be arbitrary. However, 18 of the extended LD-ranges described by Chambers *et al* overlap with our loci and from

our 29 loci, 14 are also reported by Chambers *et al* (Supplementary Table 19). Our LDSC-based h_{SNP}^2 estimate (39.8%, SE = 3.14%) is comparable with Chambers *et al* GCTA-GREML-based estimate (45.3-46.8%). The authors of ref 10 state that 732 transcripts overlap with the complete extended-LD regions of the 33 index SNPs and map 14 genes via cerebellum-specific eQTLs, whereas we map 189 genes based on <1MB genomic proximity and 32 genes via cerebellum-specific eQTLs. We further provide novel results by stratifying the heritability, defining and comparing the genetic architecture using MiXeR, and finemapping all identified loci.

The largest difference between our study and ref 10 lies within the gene-based follow up that our study provides. We perform a gene-based GWAS and identify 85 significant genes associated with cerebellar volume. We use the gene-based summary statistics for four novel gene-set analyses: cerebellar cell-type specificity analysis, temporal cerebellar gene-expression analysis, biological pathway analysis and an additional evolutionary gene-set analysis. By doing so, we show nominal significant associations between cerebellar volume gene associations on the one hand, and astrocyte specific gene-expression and prenatal cerebellar gene-expression on the other hand.

Another difference between ref 10 and our study is related to genetic correlation analyses. Chambers *et al* do not find nominal significant global genetic correlation between total cerebellar grey matter volume and psychiatric disorders (schizophrenia, bipolar, major depression, ASD and ADHD), but report an undirected pleiotropic relationship in 8 GWAS signals with a psychiatric phenotype. We added neurodegenerative disorders given the phenotypic relevance in previous literature. We report a nominal significant global genetic correlation between total cerebellar grey and white matter volume and ADHD, and directed local genetic correlations between total cerebellar grey and white matter volume and schizophrenia, Alzheimer's and Parkinson's Disease. We provide additional colocalization results to even zoom in the level of a shared causal variant between these traits."

2. *Could the authors elaborate more on how the replication sample was selected?*

We have now added details on how the replication sample was selected to the Methods section.

Methods, section "Sample", page 16, line 391

"SNP-genotypes were released for N = 488,377 participants in March 2018, with neuroimaging data available for a subsample of N = 40,682 individuals⁶⁰. From the ~20,000 subjects released in January 2020, we drew a random list of 5,000 subjects for replication purposes. From the 40,682 discovery and replication subjects available, N = 9 were excluded due to later withdrawn consent and 6,058 individuals were excluded by us because of UKB-provided relatedness (subjects with the most inferred relatives, third degree or closer, were removed until no related subjects were present), discordant sex, or sex aneuploidy. Additionally, individuals of European descent were included if their projected principal

component score was closest to and < 6 SD (based on Mahalanobis distance) from the average principal component score of the European 1000 Genomes sample ($N = 2,187$ non-European exclusions), as has been described in previous publications by our group⁶¹. Phenotype data was quality controlled as described below ($N = 513$ missing phenotypes and $N = 121$ outliers) and was matched with the quality-controlled genotype data: we continue by reporting the maximum per-SNP sample size ($N = 31,392$). After splitting subjects based on the randomly drawn replication list, we arrived at two samples: the discovery sample ($N = 27,486$) aged $M = 63.55$ ($SD = 7.52$) years with 52.50% females and the replication sample ($N = 3,906$) aged $M = 64.91$ ($SD = 7.30$) years with 54.58% females. Supplementary Table 18 gives an overview of sample quality and exclusion criteria.”

Reviewer #2

Tissink and colleagues conduct a GWAS of cerebellar volume. Overall, the GWAS is well conducted and the analysis is technically sound. The paper is well written and easy to follow.

That said, my enthusiasm is dampened by the fact that I don't think the paper is telling anything new, and seems to be analyses run through a fairly standard pipeline. Given that summary statistics for this phenotype is available already, I am left wondering what this paper is adding to the field.

We understand this comment of the reviewer, however, we still think there is novelty in our study. Our study provides the most detailed SNP- and gene-based investigation of total cerebellar volume to date and offers an interpretation of the genetic summary statistics from a wide range of perspectives. We zoom out from the variant level (annotation and gene mapping, finemapping, colocalization), to the gene level (gene-based GWAS, four gene-set analyses), to the locus level (local genetic correlations with neurodevelopmental and neurodegenerative disorders) and eventually to the genome-wide level (global genetic correlations, (stratified) heritability, polygenicity, discoverability). This is in contrast to previous studies where only GWAS summary statistics are shared (Ref 8-9), or a limited set of SNP-based follow-up analyses is provided (Ref 10). A flowchart that describes all Methods used in this revised manuscript is updated and displayed in Supplementary Figure 4. A detailed description of how these results compare and go beyond previous studies (refs 8-10) is now provided in the Supplementary Note (page 1-2) and Supplementary Table 19, please see our response to point 1 by reviewer 1.

In the updated version of the manuscript, we also modified two paragraphs of the introduction to state how our study provides novel findings to the field.

Introduction, page 2, line 38:

“One way to examine the genetic architecture of neuroimaging-derived phenotypes is to evaluate common genetic variants (single nucleotide polymorphisms or SNPs) for association in a genome-wide association study (GWAS)^{11,12}. Based on these SNP effects, cerebellar volume heritability has been estimated at 45.3 - 46.8%, and 21 genes have been identified for potential follow-up⁸. However, a large

body of research has shown that methods that exploit the polygenic signal of traits to look for convergence beyond genes onto biological pathways, cell types or developmental time windows have the potential to provide more meaningful starting points for follow-up experiments¹³. In parallel, methods that zoom in on a locus level to examine the local genetic overlap between traits or pinpoint the likely causal variant(s) facilitate the prioritization of SNPs or genes for future studies¹³. Therefore, a thorough interrogation of cerebellar volume at the level of genetic variants with detailed follow-up focused on translating genetic loci into mechanistic hypotheses could elucidate why cerebellar volume varies between individuals and contributes to neurodevelopmental and neurodegenerative disease.

We perform a GWAS on UK Biobank data of total cerebellar gray and white matter volume (discovery N = 27,486, replication N = 3,906) to assess which common genetic variants and genes contribute specifically to cerebellar volume. We annotate and finemap discovered loci to pinpoint the most impactful and likely causal variants and describe the genetic architecture of cerebellar volume by its polygenicity and discoverability compared to cerebral and subcortical volume. It is examined to what extent cerebellar volume-specific genes converge onto biological pathways or specific cell types located in the cerebellum. We further investigate whether these genes display a specific temporal expression pattern through development or associate with genes important for human evolution. The overlap between the genetic profile of cerebellar volume and neurodevelopmental and neurodegenerative disorders on a global and regional level highlights the impact of cerebellar volume genetic variation on ADHD, schizophrenia, Alzheimer’s and Parkinson’s disease. Our results provide insights into the heritable mechanisms that contribute to developing a key brain structure for cognitive functioning and mental health.”

Supplementary Information, page 6:

“Supplementary Figure 4. Flowchart of all data and methods used to obtain results as presented in this study. For more detailed information see Methods.”

1. Would the authors consider additional lines of analyses to tell us something new about the cerebellum that's not quite fully known yet using the GWAS analyses? For instance, how does the genetic architecture compare to that of the cerebral volumes/SA?

We thank the reviewer for the useful suggestion to compare the genetic architecture of the volume of the cerebellum with that of other major brain structures, such as the cerebrum. We were inspired by this suggestion and decided to extend this exploration to a third part of the brain: subcortical volume. In the updated version of our manuscript, we have added a comparison of cerebellar, cerebral and subcortical volume in terms of their genetic architecture: the number of genetic variants influencing the trait, the size of their effects on the trait and the frequency of those variants in the population. To estimate these metrics, we apply univariate MiXeR on the summary statistics of each trait. We find that cerebellar volume shows the highest discoverability (variance of non-null effect sizes) among the three volumes ($\sigma_{\beta}^2 M = 3.60 \times 10^{-4}$, $SD = 2.52 \times 10^{-5}$), meaning that it has on average stronger effects. We illustrate this effect now in Supplementary Figure 1, that shows that cerebellar volume shows the highest squared standardized effect sizes for independent significant SNPs with low minor allele frequencies (MAF). We added the following sections to our manuscript:

Methods, section “Genetic architecture definition with MiXeR”, page 19, line 464:

“In order to place our cerebellar volume GWAS into perspective, we compared its genetic architecture with the other two major volumes of the brain: cerebral volume and subcortical volume. Two SNP-based GWAS were performed for subcortical volume and grey + white cerebral volume, using the same pipeline as described for our cerebellar volume discovery GWAS described above. We used univariate MiXeR¹ (URLs) to describe the number of genetic variants contributing to each trait, their polygenicity (π : the proportion of independent causal SNPs) and discoverability (σ_{β} : the variance of the effect sizes of causal SNPs). π ranges between 0 and 1 and higher π values indicate higher polygenicity. High σ_{β}^2 indicates a high level of discoverability of causal SNPs for the trait. 1000 Genomes EUR was used as reference panel and the MHC region (chr6:26–34 Mb) was excluded from all three SNP-based GWAS summary statistics. To complete our overview of the genetic architecture per trait, we plotted the minor allele frequencies of (FUMA-defined) independent significant SNPs against their squared standardized effect sizes.”

Results, section “The SNP-based heritability of cerebellar volume is 39%: 30 genomic loci identified”, page 4, line 88:

“We estimated three metrics to describe the genetic architecture of cerebellar volume using univariate MiXeR²¹ (see Methods) and compared this to two other main volumes in the brain, namely cerebral and

subcortical volume (Supplementary Table 15). Cerebellar ($M = 4.36 \times 10^{-4}$, $SD = 4.12 \times 10^{-5}$) and cerebral volume ($M = 3.96 \times 10^{-4}$, $SD = 5.99 \times 10^{-5}$) appeared to show lower polygenicity (π ; proportion of independent causal SNPs) than subcortical volume ($M = 7.22 \times 10^{-4}$, $SD = 5.24 \times 10^{-5}$). As less polygenic traits tend to have more causal SNPs with larger effect sizes, cerebellar volume showed the highest discoverability (variance of effect sizes of causal SNPs) among the three volumes ($\sigma_{\beta^2 M} = 3.60 \times 10^{-4}$, $SD = 2.52 \times 10^{-5}$), meaning that it has on average stronger effects. This is also illustrated by Supplementary Figure S1, that shows that cerebellar volume shows the highest squared standardized effect sizes for independent significant SNPs with low minor allele frequencies (MAF). All metrics for cerebellar, cerebral and subcortical volume can be found in Supplementary Table 15.”

Supplementary Information, page 3:

“Supplementary Figure 1. Scatter plot of MAF and squared standardized effect sizes (Beta) for cerebellar, cerebral, and subcortical volume.”

2. Are there specific evolutionary signatures that can be gleaned from the analyses of the GWAS data?

The cerebellum underwent rapid expansion throughout evolution² and it is known that evolutionary cortical expansion of cognitive networks runs parallel with high expression of genes in human-accelerated regions (HAR genes)³. Upon the reviewer’s suggestion we investigated whether the mean genetic association of cerebellar volume is higher for HAR genes than for all the other genes in our

data, but we did not find direct evidence for this in our GWAS results (MAGMA analysis of HAR genes, $\beta = 9.88 \times 10^{-3}$, $SE = 2.75 \times 10^{-2}$, $p = 0.36$). We added the following sentences and paragraphs to the revised manuscript:

Methods, section “Evolutionary gene-set analysis”, page 23, line 574:

“The cerebellum underwent a significantly faster size increase throughout evolution than predicted by the change in neocortex size². We therefore investigated if any evolutionary signatures could be observed in our GWAS. For this purpose, a gene-set was curated from genes located in human accelerated regions (HARs) and used during MAGMA’s gene-set analysis. These previously identified regions are highly different between humans and other species and are suggested to have potential roles in the evolution of human-specific traits⁴. One thousand five hundred seventy-one of the HAR-associated genes were present in the cerebellar volume gene-based GWAS summary statistics. The alpha level for the HAR gene-set reaching significance was $\alpha = 0.05$.”

Results, section “Looking for convergence of cerebellar volume genes on a wide range of gene-sets”, page 8, line 188:

“We then investigated whether associated genes converged on biological pathways, an evolutionary signature, or cerebellum-specific cell types. To this end, we conducted gene-set analyses in MAGMA on the gene-based summary statistics using 7,246 MSigDB⁵ gene-sets, one gene-set including genes in human accelerated regions (HARs) that are highly different between humans and other species⁴, and a cell type-specificity analysis in FUMA using cerebellar cell types from the DropViz Level 1 database⁶. No evidence was found for enrichment in any of the tested MSigDB gene-sets (Supplementary Table 10) or the HAR gene-set (\$\beta = 9.88 \times 10^{-3}\$, \$SE = 2.75 \times 10^{-2}\$, \$p = 0.36\$ ).”

Reviewer #3

Tissink and colleagues report on a genome-wide association study on MRI cerebellar volume in 31,392 individuals included in the UK Biobank. The trait is highly heritable which led them to find 29 genomic loci that pass genome-wide significance. The genome-wide summary statistics are well-behaved and through several downstream analyses they nominate candidate genes that affect cerebellar volume. Subsequently, they study enrichment of cell-types, pathways or developmental stages. Finally, they relate cerebellar volume to neuropsychiatric traits through estimating (local) genetic correlations.

To my knowledge this is the first GWAS on cerebellar volume (besides the preprint that is cited in this paper). The main analyses are solid and these data are therefore valuable to a wide scientific audience especially involved in genetics and neuroscience.

There are some aspects of this study that could be clarified.

1. *The methods describe MRI data is available for 40,682 individuals, but eventually 31,392 are included. This means that nearly a quarter of the data is lost due to quality control. Can the authors specify at which step how many samples failed?*

The reviewer is correct in noting this difference. Individuals were excluded due to the following reasons: the released sample consisted of $N=40,682$, however 9 individuals had a later withdrawn consent and 6,058 individuals were excluded by us because of genetic relatedness and low genotyping quality. We further restricted our analyses to a European sample, excluding another 2,187 non-European subjects. Phenotypic quality control then led the exclusion of 513 subjects due to missing cerebellar volume and 121 having an outlying cerebellar phenotype. Altogether this resulted in a final dataset available for analyses of 31,794. We then reported the maximum per-SNP sample size as the definitive sample size in our paper ($N = 31,392$), which is split into $N_{\text{discovery}} = 27,486$ and $N_{\text{replication}} = 3,906$ based on a random draw of subject ID's from the latest UK Biobank neuroimaging release.

An overview of these numbers is now provided in Supplementary Table 18. We subsequently adjusted our manuscript as follows.

Methods, section “Sample”, page 16, line 391:

“SNP-genotypes were released for $N = 488,377$ participants in March 2018, with neuroimaging data available for a subsample of $N = 40,682$ individuals⁶⁰. From the ~20,000 subjects released in January 2020, we drew a random list of 5,000 subjects for replication purposes. From the 40,682 discovery and replication subjects available, $N = 9$ were excluded due to later withdrawn consent and 6,058 individuals were excluded by us because of UKB-provided relatedness (subjects with the most inferred relatives, third degree or closer, were removed until no related subjects were present), discordant sex, or sex aneuploidy. Additionally, individuals of European descent were included if their projected principal component score was closest to and < 6 SD (based on Mahalanobis distance) from the average principal component score of the European 1000 Genomes sample ($N = 2,187$ non-European exclusions), as has been described in previous publications by our group⁶¹. Phenotype data was quality controlled as described below ($N = 513$ missing phenotypes and $N = 121$ outliers) and was matched with the quality-controlled genotype data: we continue by reporting the maximum per-SNP sample size ($N = 31,392$). After splitting subjects based on the randomly drawn replication list, we arrived at two samples: the discovery sample ($N = 27,486$) aged $M = 63.55$ ($SD = 7.52$) years with 52.50% females and the replication sample ($N = 3,906$) aged $M = 64.91$ ($SD = 7.30$) years with 54.58% females. Supplementary Table 18 gives an overview of sample quality and exclusion criteria.”

2. *The methods state that 6,067 were excluded, but judging from the number more were lost ($40,682 - 31,392 = 9,290$), can the authors explain this discrepancy.*

We thank the reviewer for this point and we hope that we further clarified this issue in the point above and our updated Methods, section “Sample”, page 16 and Supplementary Table 18. To summarize, this discrepancy can be explained by further exclusions due to non-European ancestry (n= 2,187), phenotype missingness (n=513), phenotype outliers (n=121), and overlapping quality-controlled genotype and phenotype data (n = 402).

3. The authors perform LDSC to show the GWAS was overall not confounded. Personally, I'm not sure if subsequently mentioning Λ_{1000} - $\Lambda_{10,000}$ is that informative (as the N dimension is already taken into account in the LDSC h^2 estimate and intercept). Furthermore the statement that Λ increases with N is a characteristic of a polygenic trait in partially true, it is also a characteristic of a confounded GWAS that grows in size (LDSC is therefore more informative).

We agree and have removed our statements about $\Lambda_{1,000}$ – $\Lambda_{10,000}$ and instead focus solely on the LDSC h^2 estimate and intercept values in the first paragraph of our Results (section “The SNP-based heritability of cerebellar volume is 39%: 30 genomic loci identified”, page 3, line 69).

4. I do not totally understand the purpose of the out of sample prediction analyses. Is it to illustrate robustness of the reported associations? Maybe it could be combined describing the replication efforts. That r^2 for out of sample prediction is lower than h^2 is expected from theory. If the authors want to present an optimal PRS other methods than clumping and thresholding might be better (Ni et al, Biol. Psych., 2021).

We thank the reviewer for these valuable suggestions. We confirm that the purpose of our polygenic prediction analyses was to illustrate the robustness of the reported associations and we added a statement to clarify this purpose more explicitly in the Results (page 12, line 269) section. Additionally, this replication effort is now combined with the lead SNP validation section in a new paragraph (Results, section “Polygenic score prediction and lead SNP validation”, page 12, line 267) as suggested by the reviewer.

The reviewer additionally points towards a recent publication by Ni *et al* (2021) that compared nine polygenic score methods to one baseline P-value based clumping and thresholding method. These authors conclude that MegaPRS, LDpred2 and SBayesR provide significantly higher prediction statistics for schizophrenia and major depressive disorder than the P-value based clumping and thresholding method. We were unaware of this recent publication. We have now performed a parallel prediction analysis using LDpred2. Indeed, we observed an increase in prediction by 2.17%. We added the following paragraphs to our manuscript to describe this procedure and its results:

Methods, section “Polygenic scores”, page 26, line 640:

“In order to estimate how much variance in cerebellar volume could be explained by our GWAS findings, we calculated polygenic scores using the discovery summary statistics (base set) in PRSice-2⁹ and LDpred2¹⁰ (URLs). For this purpose, we randomly split the replication sample into a target set (N = 1,971) and a validation set (N = 1,983). A three-phase procedure was applied, with phase 1 representing the estimation of effect sizes in our discovery GWAS. In phase 2, SNP data (MAF > 0.1, chromosome X excluded) of the target sample was used to optimize p -value thresholding (PRSice-2) and hyper-parameters (LDpred2) to find the best fit model. We constricted our LDpred2 analyses to HapMap3 variants as recommended. All models were corrected for the same covariates as during GWAS analysis described above. During phase 3, the best model was fit on clumped SNP data of the validation set to be able to report the phenotypic variance explained and p -value unaffected by overfitting.”

Results, section “Polygenic score prediction and lead SNP validation”, page 12, line 268:

“We determined the robustness of our GWAS findings with out-of-sample polygenic score (PGS) prediction using both a classical p -value thresholding and clumping (PRSice-2⁹) as well as a Bayesian (LDpred2¹⁰) method. Hyper-parameter(s) were first optimized in a target set (N = 1,971) before the best model was applied in the validation set and a validation set (N = 1,983). As shown previously¹¹, LDpred2-based PGS explained more variance in cerebellar volume (4.53%) in our validation set than PRSice-2 (2.36%; Supplementary Table 16). It is known that the prediction by PGS can be substantially lower than the h_{SNP}^2 when GWAS sample sizes are low¹², which is often the case in imaging-based GWAS. The variance explained by PGS will eventually climb close to h_{SNP}^2 estimates if GWAS sample sizes increase, as effect sizes are then estimated with less error and differences between base and target samples will decrease¹².”

5. Can the authors report the lambda and LDSC intercept for the replication GWAS?

We have added this to the manuscript:

Results, section “Polygenic score prediction and lead SNP validation”, page 12, line 279

“The replication sample showed polygenic signal ($\lambda_{GC} = 1.029$) and was not confounded (LDSC_{intercept} = 1.009, SE = 0.007).”

6. Did the authors perform a genome-wide meta-analysis of discovery and replication dataset (given the high number of replicated loci)?

We selected a holdout sample for replication purposes shortly after the third release of UK Biobank imaging data and we did not touch this sample until all other analyses had been completed. We thus did not perform a meta-analysis of discovery and replication samples.

7. Which summary statistics are used for all downstream enrichment analyses (discovery or a meta-analysis of discovery + replication)?

The downstream post-GWAS analyses have been performed using discovery SNP- and gene-based summary statistics. We now refer to “discovery GWAS summary statistics” instead of “GWAS summary statistics” in the relevant Methods sections to clarify this.

8. figure 1c: color scale of enrichment might be easier to read on $\log_2()$ scale so it is symmetric.

We $\log_2()$ transformed the color scale of Figure 1c to enhance readability as suggested by the reviewer. Figure 1c can now be found on page 5.

Results, Section “Genomic location and functions of candidate cerebellar volume SNPs”, page 4

“Figure 1. c) Proportion of candidate SNPs with corresponding genomic location, RegulomeDB score, and brain-specific common chromatin state as assigned by FUMA. $\log_2(\text{Enrichment})$ values are colour-coded (depletion coloured in blue), corresponding p-values are available in Supplementary Table S3.”

10. For the exonic variants: it could help to report the amino-acid change and effect (SD cerebellar volume) for all of them? Can they report SE or 95% CI for the effect size?

We apologize for the inconsistency that was rightfully pointed out by the reviewer. We added Supplementary Table 4 for a complete overview of the amino-acid change and effect sizes of these exonic nonsynonymous variants. Additionally, we adapted our main text as follows.

Results, section “Genomic location and functions of candidate cerebellar volume SNPs”, page 6, line 117:

“Yet four candidate SNPs were nonsynonymous exonic SNPs (ExNS; rs1060105, rs2234675, rs13107325, rs6962772) in the *SBNO1*, *PAX3*, *SLC39A8* and *ZNF789* genes respectively, of which the latter three ExNS reached genome-wide significance (Supplementary Table 4). *PAX3* encodes for a transcription factor that is involved in neural tube and neural crest development¹³ and is known to induce axonal growth and neosynaptogenesis in the mammalian olivocerebellar tract¹⁴. The missense variant in exon 6 leads to a threonine to lysine change and each effect allele accounts for a decrease in cerebellar volume of 0.17 SD ($\beta = -1688.74$, SE = 218,27). The exonic variant in the *SLC39A8* gene causes an alanine to threonine change in the zinc transporter ZIP8 the gene encodes for. Each effect allele relates to a 0.11 SD increase in cerebellar volume ($\beta = 1157.91$, SE = 172.63). The effect alleles of rs6962772 (threonine > alanine) and rs1060105 (salcatonin > asparagine) correspond to a 0.07 SD ($\beta = 750.43$, SE = 122.01) and 0.05 SD ($\beta = 548.25$, SE = 109.13) increase respectively. The corresponding CADD scores of these four ExNS SNPs (22.9, 25.2, 23.1, 12.6 respectively) indicated high deleteriousness: a property representing reduced organismal fitness, strongly correlating with molecular functionality and pathogenicity¹⁵.”

11. line 173: not sure if MAGMA “confirms” as it is a result of an LD-informed aggregation test that includes the same SNP (and thus signal), “which is in line with” might be more appropriate.

We agree with the reviewer that it would be appropriate to reformulate the interpretation of these similar results and adapted the sentence accordingly (Results section “Top genes implicated in cerebellar volume suggest role for IGF1 regulation”, page 7, line 163).

12. The title of paragraph “Convergence of cerebellar volume genes on pathways, cell types and developmental stage” suggests the authors did find enriched pathways, cell-types and developmental stages. Instead, they describe no strong evidence of convergence on cell-types, pathways or developmental stages (after correction for multiple testing). They could consider rephrasing the title of this paragraph more neutrally.

We have changed this into: “Looking for convergence of cerebellar volume genes on a wide range of gene-sets” (page 8, line 174).

13. The BRAINSPAN enrichment analysis is interesting. If I'm correct, the MAGMA analysis treats all timepoints as individual sets, and not the temporal ordering of these sets. The enrichment in pre-natal stages is not statistically significant (after correction for multiple testing), but the trend in p-values might lead the reader to speculate. Can the authors plot the regression coefficients + SE to see if there is a consistent trend of higher/lower expression in pre-natal stages or are these coefficients all over the place? Would binning in pre-natal, childhood, puberty, adolescence and adulthood samples increase power?

As correctly stated by the reviewer, MAGMA tested the positive effect of cerebellar gene expression per timepoint in no particular order, corrected for average cerebellar gene expression across timepoints. Here, a positive effect would suggest that genes tending to have stronger genetic associations with cerebellar volume are more strongly expressed in the cerebellum. We now plotted the regression coefficients + SE per timepoint (Supplementary Figure S3b) as asked by the reviewer. This plot shows more clearly that a suggestive trend in positive and negative effects can be observed: higher prenatal and lower postnatal expression in general seem to be associated with stronger cerebellar volume gene associations. Inspired by the reviewer's comment to inspect this suggestive effect difference between pre- and postnatal gene expression, we tested the difference in mean effect between these two groups of timepoints. We used a fixed-effects model and detected a highly significant difference, indicating that the average effect of prenatal gene expression is stronger than that of postnatal gene expression. We added the following sections to our manuscript:

Methods, section "Temporal gene-set analysis", page 23, line 555:

"To determine whether there was a general difference between the effects of pre- and post-natal gene expression, we tested the difference in mean effect between those two groups of timepoints as follows. From the MAGMA output, we can model the distribution of the marginal effect estimate $\hat{\beta}_j$ of each timepoint j as $\hat{\beta}_j \sim N(\beta_j, S_j^2)$, with β_j the unknown true effect and S_j the standard error. As such, for the vector $\hat{\beta}$ of all K estimates jointly we obtain the distribution $\hat{\beta} \sim MVN(\beta, SRS)$. Here, S is a diagonal matrix containing the standard errors of all the estimates, and R is the sampling correlation matrix which we can approximate by the correlation matrix of the gene expression timepoints. We now define μ_{pre} as the mean of the β_j of pre-natal timepoints, such that $\mu_{\text{pre}} = \frac{1}{K_{\text{pre}}} \sum_{j \in \text{pre}} \beta_j$ with K_{pre} the number of pre-natal timepoints, and likewise μ_{post} as the mean of the β_j of the post-natal timepoints. These μ_{pre} and μ_{post} are estimated using the corresponding means of the $\hat{\beta}_j$. With this, we can now test the null hypothesis $H_0: \mu_{\text{pre}} = \mu_{\text{post}}$ that the means of the effects of pre- and post-natal gene expression are the same. Under this null hypothesis, the difference in estimated means $\Delta_{\hat{\mu}} = \hat{\mu}_{\text{pre}} - \hat{\mu}_{\text{post}}$ is distributed as

$\Delta\hat{\mu} \sim N(0, D^T SRSD)$, which can be used to compute a p-value. In this expression, D is a length K coding vector with a value for each timepoint, being equal to $\frac{1}{K_{pre}}$ for pre-natal timepoints and $\frac{-1}{K_{post}}$ for post-natal timepoints.”

Results, section “Looking for convergence of cerebellar volume genes on a wide range of gene-sets”, page 8, line 179:

“MAGMA gene property analysis showed that **the stronger genes were** expressed in postconceptional week 17 ($\beta = 0.018$, $SE = 0.009$, $p = 0.027$) and 35 ($\beta = 0.029$, $SE = 0.013$, $p = 0.015$), **the stronger their genetic association with cerebellar volume**. These associations did not survive Bonferroni correction (Supplementary Table 12), but a general difference between the effect direction of pre- and postnatal gene expression timepoints was apparent from Figure S3b. We used a fixed-effects model to validate this observation (Methods) and detected a highly significant difference ($p = 9.43 \times 10^{-5}$) in mean effect between prenatal ($\mu_{pre} = 0.015$) and postnatal ($\mu_{post} = -0.018$) gene expression timepoints. This indicates that, although there is no Bonferroni-significant involvement found for the gene expression at any one time point, on average the effect of prenatal gene expression is stronger than that of postnatal gene expression.”

Supplementary Information, page 5

”Supplementary Figure 3b. Results from two gene-set analysis approaches: a) Gene-expression in cell types from cerebellar mouse tissue (DropViz database in FUMA) was nominally associated with the

cerebellar volume gene-based GWAS sumstats in astrocytes, but this did not survive Bonferroni correction. b) Cerebellar gene-expression in donors from different developmental stages (Brainspan database) was nominally associated with the cerebellar volume gene-based GWAS sumstats at 17 and 35 postconceptual weeks, but these did not survive Bonferroni correction. We compared the average effect of gene-expression between pre- and postnatal timepoints, and observed a highly significant difference.”

14. The regional genetic correlation analyses highlights several loci that explain heritability in cerebellar volume and AD/PD/SCZ. An interesting question is whether this is due to two different causal variants in LD or is there an actual shared signal? This might be beyond the scope of this paper, but the authors could run colocalization analyses for this purpose. If there is indeed evidence for a shared causal signal (and not just LD), the description of gene functions in these regions are more relevant.

Our local genetic correlation (r_g) analyses indeed showed genetic similarity of cerebellar volume with schizophrenia, Alzheimer’s, and Parkinson’s Disease in four specific genomic regions. We appreciate the suggestion of the reviewer to zoom in even further to the single variant level using a colocalization method, to determine whether there is a true, causally shared genetic effect within these loci. Therefore, we ran *coloc* on the summary statistics of both traits in these four loci. We did not find evidence that both cerebellar volume and the neurodevelopmental/neurodegenerative disorder share the same causal variant in these loci, using a posterior probability threshold of 0.8.

We added the following text to our manuscript:

Methods, section “Colocalization of significant r_g loci”, page 25, line 616:

“After obtaining significant genetically correlated loci as described above, we investigated if the summary statistics support a shared causal variant for both traits. We ran colocalization analyses with the *coloc* R package¹⁶ (URLs) for the genetically correlated loci that reached significance. Coloc is a Bayesian method that provides posterior probabilities for five hypotheses (H_0 = no association with either trait; H_1 = association with trait 1, not with trait 2; H_2 = association with trait 2, not with trait 1; H_3 = association with trait 1 and trait 2, two independent SNPs; H_4 = association with trait 1 and trait 2, one shared SNP). We called the signals colocalized when the posterior probability of $H_4 > 80\%$.”

Results, section “Global and local genetic overlap between cerebellar volume and disease”, page 11, line 252:

“In addition to computing r_g within these loci, we ran colocalization analyses to determine whether cerebellar volume shared the same or a different causal variant as the neurodevelopmental and

neurodegenerative disorders in these loci (Supplementary Table 15). Given that the posterior probabilities for neither hypothesis 3 (traits have different causal variants) nor hypothesis 4 (traits share the same causal variant) reached a convincing threshold (80%), we cannot conclude how the genetic signals of cerebellar volume and the neurodevelopmental and neurodegenerative disorders within these loci relate on a single-variant level.”

15. The label on y-axis of fig 4 is a bit unclear, specifically the difference between ρ and r_g , maybe better to write P and r_g ? And clarify that labels in the plot correspond to r_g estimates.

As suggested by the reviewer, we changed the y-axis label accordingly and added “ r_g =” to every label in the plot to clarify that the numbers correspond to r_g estimates. The improved figure can be found on page 11.

“Figure 4. Local genetic correlations (r_g) between cerebellar volume and neurodevelopmental and neurodegenerative disorders. The red line indicates the significance threshold, Bonferroni corrected for the number of loci tested in all seven traits. As during global r_g , we meta-analysed the ADHD, ASD and SCZ summary statistics to represent the genetic signal of the overarching neurodevelopmental disorder dimension and similarly meta-analysed PD and AD summary statistics to capture the genetic signal of the neurodegenerative disorder dimension (see Methods).”

16. The authors rightfully acknowledge that it is hard to pinpoint the genetic correlations to cerebellar volume, as itself is correlated to the volume of other brain regions. Could multivariate Mendelian Randomisation studies help here? Of course, this could be beyond the scope of this study.

We thank the reviewer for acknowledging the challenge of specificity that the imaging-genetics

community is facing. How the genetic architecture of cerebellar volume overlaps with the volume of other brain regions was also pointed out by reviewer 2. Therefore, we now provide a characterization of cerebellar volume genetic architecture and how this corresponds to cerebral and subcortical volume. The use of multivariate Mendelian Randomization is beyond the scope of this study, and therefore we would like to point the reviewer to our response on comment 1 by reviewer 2.

17. Thanks for sharing the summary statistics upon publication, will they be deposited in the GWAS catalog?

We will contact GWAS Catalog to share summary statistics upon publication, next to sharing them on our department website https://ctg.cnr.nl/software/summary_statistics/.

REVIEWERS' COMMENTS:

Reviewer #2 (Remarks to the Author):

I thank the authors for a thorough response to my comments. I particularly appreciate the fact that they took my fairly broad comments in cheerful spirits and ran additional analyses.

I have two further comments, emerging from your miXer findings of cerebellar, subcortical, and cerebral volumes.

1. Can you define what you mean by subcortical volume? There are several subcortical structures, did you combine the volume of all of them? If yes, what's the rationale for that? If no, can you please colour code the subcortical structures separately in Supplementary Figure 1?
2. I find the miXer results intriguing. Would the authors be happy to follow up on this with bivariate genetic correlations and local genetic correlations to dig a bit deeper into this, and understand to what extent the genetic findings differ? The subcortex and cerebellum have had different evolutionary pressures compared to the cortex and this will be interesting.

Reviewer #3 (Remarks to the Author):

The authors have answered my questions and presented some interesting new analyses.

Please find below our point-by-point response to the comments raised by reviewer #2 together with the adjustments we made to the manuscript. The questions and comments of the reviewer are presented in *italics*, our response in normal typeset. We marked changes to the text in blue in the updated version of our manuscript.

Reviewer #2

I thank the authors for a thorough response to my comments. I particularly appreciate the fact that they took my fairly broad comments in cheerful spirits and ran additional analyses. I have two further comments, emerging from your miXeR findings of cerebellar, subcortical, and cerebral volumes.

1. Can you define what you mean by subcortical volume? There are several subcortical structures, did you combine the volume of all of them? If yes, what's the rationale for that? If no, can you please colour code the subcortical structures separately in Supplementary Figure 1?

We would like to thank the reviewer for their positive evaluation of our response and additional analyses. We confirm that the subcortical volume phenotype as used in GWAS and subsequently in MiXeR analyses represents a sum of subcortical volumes. Specifically, we used the UK Biobank field-ID 26517 for SubCortGray, which includes a voxel count of structures identified as subcortical gray matter (thalamus, caudate, putamen, pallidum, hippocampus, amygdala, accumbens, ventral diencephalon, substantia nigra) and excludes the brain stem and cerebellum. The rationale for this is to provide the reader with a comparative overview of the genetic architecture of cerebellar volume and the two other major structures of the brain: the cerebrum and the subcortex. We have updated our manuscript to state this clearer now:

Methods, section “Genetic architecture comparison with cerebral and subcortical volume”, page 25, line 613

In order to place our **total** cerebellar volume GWAS into perspective, we compared its genetic architecture with the other two major volumes of the brain: **total** cerebral volume and **total** subcortical volume. Two SNP-based GWAS were performed for **total** subcortical volume (**thalamus, caudate, putamen, pallidum, hippocampus, amygdala, accumbens, ventral diencephalon, and substantia nigra**) and **total** grey + white cerebral volume, using the same pipeline **and covariates** as for our cerebellar volume discovery GWAS described above.

Results, section “The SNP-based heritability of cerebellar volume is 39%: 30 genomic loci identified”, page 4, line 90

We estimated three metrics to describe the genetic architecture of **total** cerebellar volume using univariate MiXeR¹⁷ (see Methods) and compared these to two other **major** volumes in the brain, namely

total cerebral and total subcortical volume (Supplementary Data 2). Cerebellar ($M = 4.36 \times 10^{-4}$, $SD = 4.12 \times 10^{-5}$) and cerebral volume ($M = 3.96 \times 10^{-4}$, $SD = 5.99 \times 10^{-5}$) appeared to show lower polygenicity (π ; proportion of independent causal SNPs) than subcortical volume ($M = 7.22 \times 10^{-4}$, $SD = 5.24 \times 10^{-5}$). As less polygenic traits tend to have more causal SNPs with larger effect sizes, cerebellar volume showed the highest discoverability (variance of effect sizes of causal SNPs) among the three major brain volumes (σ_{β}^2 $M = 3.60 \times 10^{-4}$, $SD = 2.52 \times 10^{-5}$), meaning that it has on average stronger effects. This is also illustrated by Supplementary Figure S1, that shows that cerebellar volume shows the highest squared standardized effect sizes for independent significant SNPs with low minor allele frequencies (MAF). All metrics for cerebellar, cerebral and subcortical volume can be found in Supplementary Data 2. Genetic correlations between these three major brain volumes are described in Supplementary Note 3.

2. I find the miXer results intriguing. Would the authors be happy to follow up on this with bivariate genetic correlations and local genetic correlations to dig a bit deeper into this, and understand to what extent the genetic findings differ? The subcortex and cerebellum have had different evolutionary pressures compared to the cortex and this will be interesting.

We followed-up on the suggestion of the reviewer and added both global and local genetic correlations between cerebellar, cerebral, and subcortical volume.

Methods, section “Genetic architecture comparison with cerebral and subcortical volume”, page 25, line 614

In order to place our cerebellar volume GWAS into perspective, we compared its genetic architecture with the other two major volumes of the brain: cerebral volume and subcortical volume. Two SNP-based GWAS were performed for subcortical volume (thalamus, caudate, putamen, pallidum, hippocampus, amygdala, accumbens, ventral diencephalon, and substantia nigra) and grey + white cerebral volume, using the same pipeline and covariates as for our cerebellar volume discovery GWAS described above. First, we used univariate MiXeR¹⁷ (URLs) to describe the number of genetic variants contributing to each trait, their polygenicity (π : the proportion of independent causal SNPs) and discoverability (σ_{β}^2 : the variance of the effect sizes of causal SNPs). π ranges between 0 and 1 and higher π values indicate higher polygenicity. High σ_{β}^2 indicates a high level of discoverability of causal SNPs for the trait. 1000 Genomes EUR was used as reference panel and the MHC region (chr6: 26–34 Mb) was excluded from all three SNP-based GWAS summary statistics. To complete our overview of the genetic architecture per trait, we plotted the minor allele frequencies of (FUMA-defined) independent significant SNPs against their squared standardized effect sizes. Second, we used LDSC and SUPERGNOVA to compute global and local r_g between cerebellar, cerebral and subcortical

volume. The alpha level for significance was Bonferroni corrected for the number of traits (global: $\alpha = 0.05 / 3 = 1.66 \times 10^{-2}$) and loci investigated (local: $\alpha = ((0.05/2,257 \text{ loci})/3) = 7.38 \times 10^{-6}$).

Supplementary Note 3, section “Genetic correlations cerebellar volume with cerebral and subcortical volume”, page 2, line 62

We followed up on comparing the genetic architecture of cerebellar volume with cerebral and subcortical volume by estimating global (Supplementary Data 20) and local (Supplementary Data 21) genetic correlations (r_g). Cerebellar and cerebral volume were significant genetically correlated ($r_g = -0.47$, SE = 0.04, $p = 1.35 \times 10^{-26}$). Note that, by correcting the cerebral and cerebellar volume GWAS for total brain volume, we correlated SNP effects between relative volumetric phenotypes. The negative r_g sign is expected from this correction and opposing effect directions can't be interpreted in an absolute volumetric manner. On a locus level, we observed a significantly correlated locus after Bonferroni correction ($p < 7.38 \times 10^{-6}$) between cerebellar and cerebral volume on chromosome 12 (101,857,796-102,961,329, corr = -1.07, $p = 3.13 \times 10^{-8}$). Interestingly, a neighboring locus on chromosome 12 (BP 99,816,842-101,857,521) showed significant r_g between subcortical and cerebellar volume (corr = -1.03, $p = 2.00 \times 10^{-9}$). These correlated loci overlap with a genome-wide significant locus identified in this study and may contain variants with distributed effects on the volume of the three major brain structures, whereas the other identified loci seem more cerebellar volume-specific. Moreover, we did not find global r_g significantly different from zero between subcortical volume and either cerebellar ($r_g = 0.03$, SE = 0.05, $p = 0.59$) or cerebral volume ($r_g = -0.08$, SE = 0.05, $p = 0.09$), suggesting a more distinct genetic architecture of subcortical volume compared to cerebral and cerebellar volume.